# Desegregation of neuronal predictive processing

Bin Wang [1,2], Nicholas J. Audette[3], David M. Schneider[3] &
Johnatan Aljadeff [4] ✉

Neural circuits construct internal 'world-models' to guide behavior. The predictive processing framework posits that neural activity signaling sensory predictions and concurrently computing prediction-errors is a signature of those internal models. To understand how the brain generates predictions for complex sensorimotor signals, we investigate the emergence of high-dimensional, multi-modal predictive representations in recurrent networks. Contrary to previous proposals of functionally specialized cell-types, stimulus and prediction-error representations are desegregated in networks performing robust predictive processing. We confirmed these model predictions by using a rich stimulus-set to violate animals' learned expectations. We propose that predictive processing is optimal when excitation/inhibition balance is loose, and reveal distinct functional roles of excitatory and inhibitory neurons. Together, we demonstrate that neural representations of internal models are highly distributed, yet structured to support flexible readout of behaviorally-relevant information. Our results advance the understanding of how internal models are computed, by incorporating different computations into a unifying model.

Predictive processing—computing the expected values of sensory, motor, and other task-related quantities—is thought to be a fundamental operation of the brain[1,2]. Violation of internally generated expectations, known as prediction-errors, is an important neural signal that can be used to guide learning and synaptic plasticity[3,4]. Signatures of predictive processing, including neural correlates of prediction-errors, were identified in multiple brain circuits, and across animal species[2,5–7]. Two well-studied experimental paradigms for predictive processing are motor-auditory[8–13] and visual-auditory predictions[14–16] in the mouse cortex. Previous work has proposed that a canonical cortical microcircuit underlies the computation of predictions and prediction-errors[2,8,9,17–19]. While some predictions of this proposed microcircuit were confirmed in specific scenarios, the hypothesis that this circuit-motif serves as a general mechanism for predictive processing faces a number of challenges.

First, typical experimental paradigms study predictive processing in animals trained to make a single association[12,16,20], while natural sensorimotor associations are typically high-dimensional (e.g., speech production[21]), as well as context-dependent[22,23]. Little is known about how specific neural architectures in the brain learn to implement such high-dimensional multimodal computations. Second, predictive processing of multimodal inputs was also observed in multiple brain circuits outside of the mammalian cortex, including subcortical circuits mediating placebo analgesia (prediction-based suppression of pain[24]); and motor-visual circuits in cephalopods that predict the animal's appearance to an external observer, and use it to generate high-dimensional camouflage patterns[25]. It is unknown whether these neural circuits employ similar or entirely different strategies for computing multimodal predictions compared to those in the mammalian cortex[2,8,9,17,19]. Third, predictive neural representations emerge on

[1]Department of Physics, University of California San Diego, La Jolla, CA, USA. [2]Mortimer B. Zuckerman Mind Brain Behavior Institute, Department of Neuroscience, Kavli Institute for Brain Science, Columbia University, NY New York, USA. [3]Center for Neural Science, New York University, New York, NY, USA. [4]Department of Neurobiology, University of California San Diego, La Jolla, CA, USA. ✉e-mail: aljadeff@ucsd.edu

timescales ranging from ~ 1 min[26,27], ~ 1 h[28,29], to days[16,30]. This suggests that predictive processing is supported by plasticity mechanisms operating on a range of timescales (including short-term plasticity[31]), and that circuit reorganization may not always be required for implementing predictive computations.

Evidence that sensory processing is strongly modulated by sensory predictions has motivated many theoretical studies. Classical literature proposed that predictive coding may explain visual receptive field properties when animals are presented with natural images[1,32–35]. These nonlinear autoencoder-type models can be interpreted as inferring the latent causes of sensory inputs, aligning with a Bayesian perspective[36–39]. Subsequent work has extended these models to account for neural activity correlated with prediction-errors, highlighting the role of excitation-inhibition (E/I) balance in generating such signals[40–47]. However, these studies typically focus either on multimodal predictions involving only a small number of stimulus-pairs, or on high-dimensional predictions within a single sensory modality. In addition, most comparisons have been made against coarse-grained electrophysiological or neuroimaging data[7,48]. Despite the emerging experimental data on the roles of motor actions[2,11,20,49,50] and cross-modal interactions[16] in shaping sensory representations, we still lack cellular-level and circuit-level understanding of neural mechanisms underlying these multi-modal predictive computations. This limits our ability to test hypotheses related to the underlying circuit computations based on modern large-scale neural recordings.

Another major gap from both experimental and modeling perspectives is multimodal predictive processing in high-dimensions: (*i*) What are the neural representations of predictable and unpredictable sensory and motor variables in natural conditions, where stimuli form rich ensembles with complex inter-dependencies potentially spanning multiple sensorimotor modalities[7,51,52]? (*ii*) What are the circuit mechanisms underlying the computation of those representations, and how are they shaped by experience and stimulus complexity (e.g., dimensionality) in the environment? Specifically, it remains unknown whether circuits that implement predictive processing of multimodal high-dimensional stimulus ensembles are functionally segregated[1,2,17,18], and if so, whether this segregation emerges through learning or depends on cell-types with distinct molecular markers.

We address these questions by examining the predictive representations in recurrent networks processing multimodal high-dimensional inputs during and after learning, and by relating this model to cellular- and population-level neural recordings. Motivated by recent experiments[9,11,12], we focus on multimodal associative learning paradigms where the network learns to associate multiple uncorrelated stimulus-pairs from different sensorimotor modalities. We adapted the classical predictive coding framework to these paired multimodal inputs and investigated the neural responses within a simplified network model. From a mechanistic perspective, we provide novel predictions on the expected degree of excitation/inhibition balance in the high-dimensional regime, and shed light on the role that E/I balance plays in canceling interference between multiple learned stimuli. Moreover, since E/I balance is enforced by mechanisms operating on heterogeneous timescales[53], our work will allow incorporating seemingly unrelated phenomena into a unifying model, e.g., predictive responses that change as a result of short- or long-term plasticity. From a functional perspective, the model suggests that predictive processing of multimodal high-dimensional stimuli is robust when the representations of stimuli and prediction-errors are desegregated at the cellular-level. Finally, we extend these results to examine the distinct roles played by excitatory and inhibitory neurons in generating internal predictions and to assess the dynamical and layer-specific multimodal predictive representations. These results provide insights for how different architectures and biological mechanisms support multimodal, high-dimensional predictive processing in sensory brain regions, and serve as further empirical validation of our modeling.

The analysis and results we present here extend previous studies on predictive processing to multimodal high-dimensional sensorimotor inputs and generate novel predictions that we confirmed based on experimental data. Therefore, we believe that our work reveals principles of predictive processing across species and brain regions in naturalistic environments and provides a quantitative framework for design and analysis of future experiments to decipher neural circuits underlying those computations.

## Results

### Recurrent networks that learn to generate high-dimensional predictions

We studied the neural representations formed in recurrent neural networks that perform predictive processing of multi-modal sensory and motor inputs. We focused on a typical associative training scenario where animals are presented with pairs of sensory stimuli simultaneously[9,11,12] or after a short delay[16]. The stimuli comprising each pair are typically of different sensory modalities (e.g., auditory-visual[16]), or involve a sensory-motor association (e.g., auditory-locomotion[12]). Each component of the association–a stimulus or motor action–is separately familiar to the animal. Thus, in this scenario, we can specifically study predictive computations which associate these components with each other, and are learned over time through synaptic-weight updates[9,11,12,16,20,49]. The network model consists of a large number $N$ of recurrently connected neurons whose firing-rates depend nonlinearly on the input current driving their responses (Fig. 1a). The presentation of the $k$-th stimulus-pair to the network corresponds to the values of the components $x_k$ and $y_k$ of the stimulus input vectors $x$, $y$. We considered scenarios with $P$ stimulus-pairs, so $x$, $y$ are $P$-dimensional input vectors. The strength of the $k$-th stimulus input to each neuron corresponds to the components of the stimulus-specific feedforward synaptic weight vectors $w_k$ and $v_k$, both of which are $N$-dimensional vectors. When $P$ is of the same order as the number of neurons $N$, the network is said to perform high-dimensional predictive processing. According to the above definitions, different stimulus-pairs are orthogonal to each other, and the number of stimulus-pairs $P$ is equal to the stimulus dimensionality. The term "high-dimensional" refers to the fact that the subspace spanned by all the stimulus input vectors has large dimensionality. This simplifying assumption, that there are no correlations across stimulus pairs, is made for mathematical tractability.

Before training, the feedforward weight vectors corresponding to each stimulus-pair are random and uncorrelated within the pair (i.e., $w \cdot v = 0$, we dropped the subscript $k$ for convenience). During training, those weights become correlated ($w \cdot v = \mu$, with $\mu > 0$), consistent with measurements of learning-induced functional reorganization of excitatory synaptic connections[54–56]. These weight changes of recurrent connections are chosen to minimize errors between internally generated predictions and the actual stimuli, while maximizing the overall encoding efficiency (Methods). This network model extends previous autoencoder-type models of predictive processing[1,33–35,39–41,43–47] to incorporate multimodal predictions and is consistent with the notion that predictive coding is a signature of Bayesian inference of latent variables representing the state of the world[57–59] (Methods). Under these assumptions, we derived analytical expressions for the key statistics of neural activity in the network for different stimulus inputs, at different stages of learning. Notably, in the large-network limit, these population-level statistics are independent of the randomness in the initial weights (Methods). The resulting neural activity enables the flexible reading-out of the stimulus identity, predicting the "missing" stimulus (i.e., predicting $y$ based on $x$), and evaluating the prediction-error (Fig. 1a). We applied these results (SI §2–3) to investigate the structure of multi-modal predictive neural representations and the circuit mechanisms supporting it.

We first examined neural responses during learning in the *match* ($x = y$), and *mismatch* ($x \neq y$) conditions. We set $x$ and $y$ to be binary

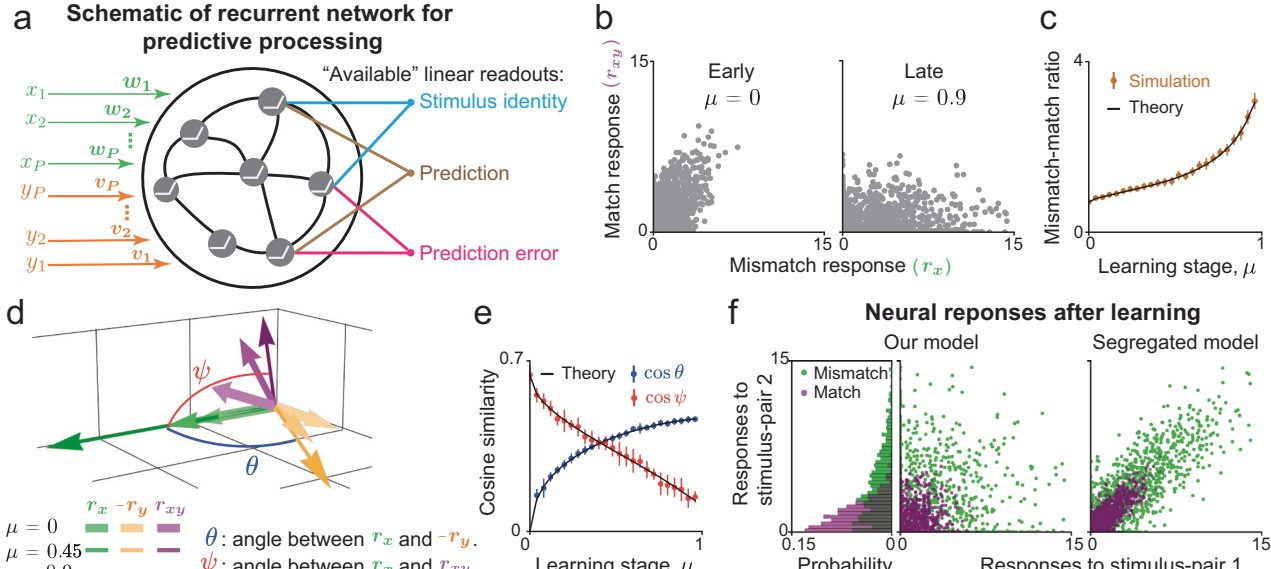

**Fig. 1 | Emergence of predictive stimulus representations in a recurrent network model during learning. a** Schematic of a recurrent network model driven by $P$ pairs of stimuli ($x$ and $y$). Associative training increases the correlations between the feedforward weights carrying the input signals ($\boldsymbol{w}$ and $\boldsymbol{v}$). The recurrent weights jointly optimize prediction-errors and overall encoding efficiency. The neural representation formed under such optimal recurrent connectivity allows reading-out the identity of the presented stimulus, predicting a "missing" stimulus, and evaluating the prediction-error. **b** Firing-rate responses of individual neurons in the match and mismatch conditions. Initially match and mismatch responses are correlated. After learning, responses are less correlated, and match responses are suppressed while the mismatch responses are amplified. **c** The ratio between average firing-rates in the mismatch and match conditions increases during learning. **d** Reduced three-dimensional neural activity space. Each vector represents the mean-subtracted firing-rate vector of neurons in the network at different conditions and stages of learning. **e** Learning leads to anti-correlation between neural responses to the stimuli $x$ and $y$ when presented separately (blue), and decorrelates the neural responses in the match and mismatch conditions (red), quantified by the angle between the population vectors. **f** Firing-rate responses of individual neurons to two stimulus-pairs in the match and mismatch conditions. In the network model studied here (left), there are no correlations between the responses to the two stimuli. Those responses are expected to be strongly correlated in a model in which predictive coding is functionally segregated (right). Circles and error-bars in (**c**, **e**) correspond to mean ± 1 S.D. computed over $n = 10$ instances of the network in Eq. (1). See Methods for additional details.

variables corresponding to the presence ($x$, $y = 1$) or absence ($x$, $y = 0$) of visual-auditory, visual-motor (V-M), or auditory-motor (A-M) pairings[12,13,16,20]. Our mathematical formalism extends to scenarios where more than two stimuli are predictive of each other, and where the inputs to the network vary continuously (e.g., running- or visual-flow-speed[20,49]; Methods). Before associative training ($\mu = 0$), most of the neurons in the network have comparable match ($r_{xy}$) and mismatch ($r_x$, $r_y$) responses (Fig. 1b). After training ($\mu = 0.9$), match responses are suppressed while mismatch responses are amplified (Fig. 1b). Correspondingly, the ratio of average mismatch and match firing-rates increases (Fig. 1c), consistent with associative learning experiments[12,16,20]. Thus, the presence of stimulus $y$ suppresses the response evoked by stimulus $x$, and generates a prediction (or expectation) of $x$. Amplified mismatch responses are interpreted as prediction-errors[2,7].

During learning, the mismatch responses ($r_x$, $r_y$) become anti-correlated (Fig. 1d, e), i.e., the presence of stimulus $y$ more effectively suppresses responses to $x$ alone. This anti-correlation does not appear between $r_x$ and $r_y$ of another stimulus-pair (Fig. S1a), suggesting that the predictive signal triggered by stimulus $y$, is specific to its paired stimulus $x$, consistent with refs. 12,60. The specific suppression of responses to predictable stimuli is accompanied by a weaker, global gain that depends on the overall magnitude of sensory input (SI §3). Furthermore, match and mismatch neural responses decorrelate during learning (Fig. 1d, e), consistent with ref. 16, suggesting that neural responses can be used to distinguish between presentation of stimulus $x$ in the match or mismatch condition. Notably, owing to the neural response nonlinearity, the match response is not a sum of the two mismatch responses ($r_{xy} \neq r_x + r_y$, Fig. 1d).

Next we examined neural responses when the network is trained with two stimulus-pairs ($P = 2$, Fig. 1f), making a step towards the high-

dimensional scenario. Refs. 2,17,18,38,58 proposed that neurons involved in predictive processing are functionally segregated, i.e., neurons that signal prediction-error for one stimulus association tend to signal prediction-error for other associations, and similarly for "representation" neurons that encode the stimulus itself. This proposal would predict a high degree of correlation between neural responses to two stimulus-pairs (Fig. 1f, right). However, we found no such correlation in the recurrent network model we considered here (Fig. 1f, left). This implies, for example, that a neuron that signals prediction-error for stimulus-pair 1, may have a selective response to stimulus $x$ "itself" for pair 2, and raises the question of what circuit mechanisms may support this cellular-level desegregation of response types.

## Learning and stimulus dimensionality determine the properties of effective predictive processing circuits

We then investigated circuit mechanisms underlying multi-modal high-dimensional predictive processing. We decomposed the input to each neuron into feedforward and recurrent components, which respectively correspond to the actual stimulus signal and to internally generated predictions (Fig. 2a), similarly to analyses of previous experiments[2,12,17,20]. To quantify the relative contribution of each component, we follow the excitatory/inhibitory (E/I) balance literature[41,61], and define the balance level $B$ as the ratio between the total feedforward input and the net input to each neuron, in each condition (Fig. 2a). We show in a later section that the balance level $B$ is closely linked to the degree of E/I balance by extending the model to include separate excitatory and inhibitory populations.

During associative learning, internally generated predictions become more accurate, facilitating more robust cancellation of the feedforward stimulus input by recurrent feedback conveying prediction signals. Thus, the overall balance level increases in the match

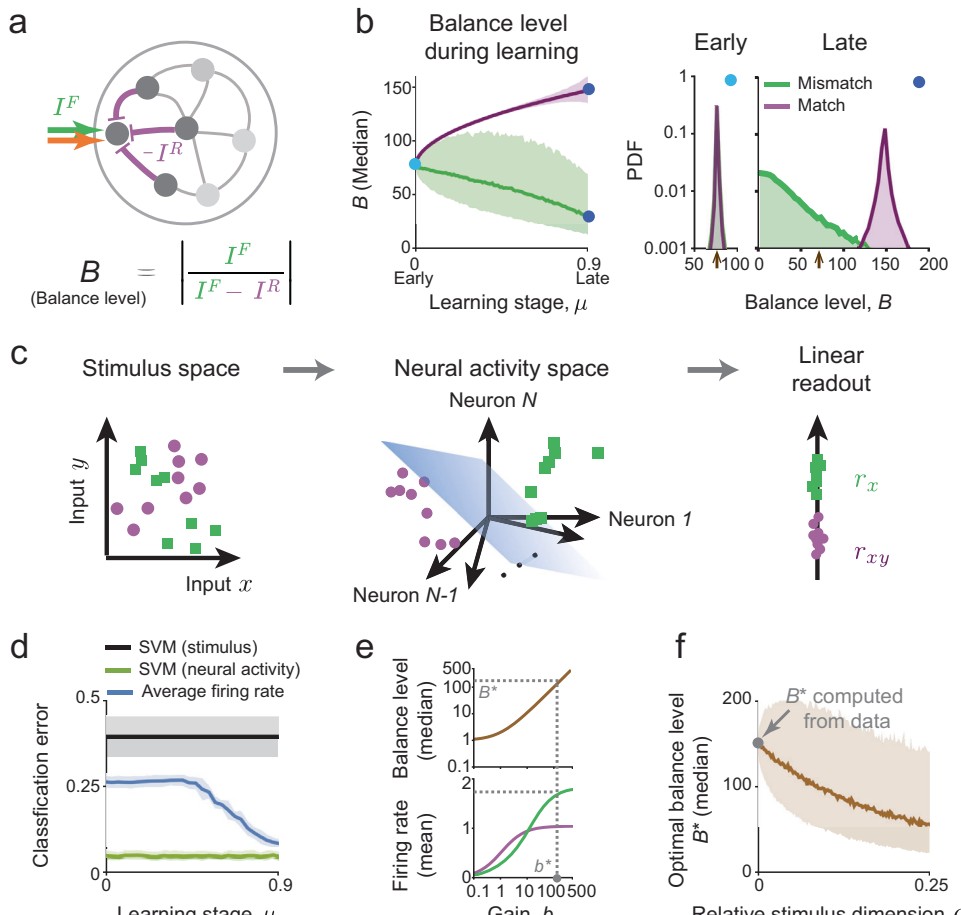

**Fig. 2 | Balance between feedforward and recurrent inputs supports predictive processing. a** The input to each neuron is decomposed into feedforward and recurrent components, which respectively correspond to the actual stimulus signal and internally generated predictions. Each neuron's balance level $B$ is the ratio between the total feedforward input and the net input (Methods). **b** The median of $B$ in the match and mismatch conditions during learning [left, shaded area represents the inter-quartile-range (IQR)]. "Snapshots" of the distributions of $B$ early and late in learning show that the distributions become separable in match and mismatch conditions (right). The arrows on the $x$-axis indicate the distribution mode early in learning. **c** Schematic showing the nonlinear transformation from the stimulus space (left) to neural activity space (center), which facilities a linear readout of relevant stimulus features (here, decoding if $x$ is presented in the match/mismatch condition). **d** Error of a support vector machine (SVM) classifier trained to identify the match/mismatch condition based on the input (black) and on neural responses (green). After learning, a linear classifier based on the average firing-rate (blue) performs almost as well as the optimal classifier. Data are presented as mean ± 1 S.D. computed over $n = 10$ repeats. **e** Illustration of the procedure to determine the optimal $b^\star$. Increasing $b$ leads to a larger margin between match and mismatch responses (improved separability) at the cost of higher firing-rates (bottom). The optimal balance level $B^\star$ is determined by constraining the average firing-rate in the mismatch condition and minimizing it in the match condition. **f** Increasing the stimulus dimension leads to decrease in $B^\star$, i.e., a more loose balance (line: median; shaded area: IQR). At $\alpha = 0$, we fit $B^\star$ to experimental data (Methods, ref. [20]).

condition but decreases in the mismatch condition (Fig. 2b, left). Such changes to the overall balance level have been suggested as the mechanism underlying global neural responses to "rare" stimuli[40–42]. Notice that the balance level distributions (over neurons and stimuli) are initially similar in the match and mismatch conditions, but become significantly different in late stages of learning (Fig. 2b, right). Indeed, after learning, the mode of the balance level distribution is at $B \approx 0$ in the mismatch condition, which explains the strong prediction-error responses.

To understand the role of balance in a scenario of multimodal predictive processing, we examined its effect on the nonlinear transformation the network performs, from input stimuli to neural activity (Fig. 2c). Although the network connectivity is optimized for generating multimodal predictions (rather than prediction-errors; Methods), the geometry of neural responses after training facilitates robust readout of prediction-errors. Specifically, while prediction-errors cannot be read-out by a linear decoder from the stimulus input, such a readout is feasible once the input is nonlinearly transformed into the network's high-dimensional response (Fig. 2d; Methods). Moreover,

while the prediction-error itself is stimulus-specific, the decoder that performs this computation can be stimulus-independent after learning: a decoder that uses only the average firing-rate has similar performance to the optimal stimulus-specific Support Vector Machine decoder (Fig. 2d), suggesting that the learned geometry of neural responses enables applying the same decoder to all stimulus-pairs without "re-learning".

Given the essential role of the nonlinear transformation for predictive processing, we next focused on the effect of the overall nonlinear gain parameter $b$ (Methods, ref. [43]). We found that increasing $b$ leads to increases of the average match and mismatch firing-rate responses, together with a wider margin between them (Fig. 2e, top). Therefore, large $b$ facilitates decoding prediction-errors, at the cost of increased overall neural activity. Motivated by this observation, and since $b$ is an intrinsic network quantity that can potentially be adjusted dynamically, we sought to find an optimal value (denoted $b^\star$). Specifically, we constrained the average network response in the mismatch condition to be larger than a certain threshold, while requiring a minimal but nonzero average response in the match condition

(Fig. 2e), consistent with reports of weak neural responses to predictable stimuli[12,20]. The resulting $b^*$ corresponds to an optimal balance level $B^*$ supporting efficient encoding and robust decoding (Fig. 2e, bottom).

We carried out this optimization procedure for networks trained to perform predictive processing of stimulus ensembles with increasing dimensionality (i.e., increasing $\alpha = P/N$), with the same firing-rate constraints chosen such that the value of $B^*$ at $\alpha = 0$ matches experimental data. We additionally assumed that an "over-trained" animal learns a single stimulus-pair (i.e., $\alpha = 1/N \approx 0$). Surprisingly, we found that the optimal balance level decreases with $\alpha$ (Fig. 2f), an effect which is robust to noise and to changes in the stimulus statistics (Fig. S1b, c). This decrease in balance acts to limit the interference between internally generated predictions corresponding to the larger number of stimulus-pairs (Methods). We therefore expect networks performing predictive processing in natural conditions (large $\alpha$) to exhibit "loose" balance, which minimizes the overall effect of interference arising from learning to generate a large number of internal predictions.

We used neural activity recorded from animals trained on visual-motor (V-M)[20] and auditory-motor (A-M) associations[12] to constrain our network model. Specifically, we assume that the neural circuit in the corresponding sensory region operates at the optimal balance level. By leveraging the relation between the neural activity in the network and the balance levels in the network model after training, we estimated the balance levels in the mouse sensory cortex (Methods). In the V-M experiment[20], mice were trained to associate their running speed with the speed of visual-flow in virtual reality (Fig. 3a). The voltage of primary visual cortex neurons was intracellularly recorded in the match and mismatch conditions. Fitting the average voltage change in the two conditions to the model gives the estimated balance level $B^*_{V-M} = 162 \pm 61$. A consistent result was obtained in the A-M experiment[12], where mice were trained to press a lever and received closed-loop auditory feedback (Fig. 3b, c). Here, the recording was extracellular, so fitting $B^*$ relied on a slightly modified procedure (Methods).

Notably, balance level estimates were consistent across animals (Fig. S2); and laboratories (Fig. 3), despite the fact that the experiments studied different brain regions and sensory modalities, using different methods. While these factors may affect the balance level to some degree, our analysis predicts that the balance level can decrease by up to one order of magnitude when the stimulus dimension increases (Fig. 2e and Fig. S1b, c). This prediction could be confirmed if future experiments reveal a more loose balance in animals habituated to rich sensory environments.

## Stimulus and prediction-error representations are desegregated in the model

We next investigated how different functional responses are organized within the network. Previous work postulated that two distinct neural populations exist in predictive processing circuits: (i) internal representation (R) neurons that "faithfully" represent external sensory stimuli and encode internal predictions, and (ii) prediction-error (PE) neurons, which signal the difference between the actual stimulus inputs and internal predictions. Given that neurons selective to these signals also exist in our network model, we wondered whether they form functionally segregated populations. We adopted classification criteria used in experimental work (Methods, refs. 2,60): R neurons are those which respond strongly and similarly in match and mismatch conditions, while PE neurons are those which respond strongly in the mismatch condition but weakly in the match condition (Fig. 4a). We note that these definitions of R and PE neural responses are stimulus-specific: the same neuron may respond differently to different stimulus-pairs. We further note that this definition for PE neuron used in the experimental work[2,60] differs from that

used in classical predictive coding studies[1,17,37]. In the latter, "predictive error neuron" refers to specific computational units dedicated to comparing top-down predictions to bottom-up sensory inputs. We adopt this experimental definition of PE neuron to connect the modeling results to experimental data and illustrate the differences in the prediction error computation between the recurrent network model studied here and models with dedicated populations of error neurons. However, the two definitions are related: a dedicated population of "error" neurons, as hypothesized in classical works[1,17], would always be classified as PE neurons by this experimental definition and would never be classified as R neurons for any stimulus-pair.

Based on these criteria, we first computed the fractions of R and PE neurons when the network learns a single stimulus association ($P = 1$, Fig. 4b). As training progresses, the fraction of PE neurons increases significantly, consistent with experiments[16,62], and with the notion that the network learns to "recognize" the stimulus pairing. This result is independent of the classification criterion (Fig. S3). The fraction of R neurons remains unchanged (Fig. 4b), though we note that the trend does depend on the criterion (Fig. S3).

We next asked how neurons responded to more complex stimulus ensembles, specifically for two learned pairs of stimuli. The hypothesis that predictive processing is segregated[2,18] asserts that if a neuron is a PE neuron for stimulus-pair 1, and if it is active during presentation of stimuli from pair 2, it will likely be categorized as a PE neuron with respect to those stimuli too. To test this hypothesis, we computed the joint distribution of neural responses in the four relevant conditions (mismatch/match × stimulus-pair 1/2) and categorized each neuron as R or PE, separately for each stimulus-pair (Methods). We started with the low-dimensional scenario, where the two stimulus-pairs in question are the only stimuli learned by the network ($P = 2$, $\alpha = P/N \approx 0$). Surprisingly, under the data-constrained parameters, although many neurons belong to the same functional type with respect to the two stimulus-pairs, approximately 25% of neurons are in fact mixed: they are classified as having different functional types across the two stimulus-pairs (Fig. 4c, left).

Furthermore, increasing the dimension of the stimulus the network learns, leads to a twofold increase in the fraction of mixed neurons (Fig. 4c, d). Intuitively, loose balance between high-dimensional feedforward and recurrent inputs leads to a broad balance level distribution across the network (Fig. S4a). That broad distribution, in turn, affords each neuron flexibility to encode different features for different stimulus-pairs. The fraction of mixed neurons shown in Fig. 4d corresponds to two specific stimulus-pairs. When we considered instead the entire learned stimulus-set, most of the neurons are mixed with respect to at least two pairs (Fig. S4b). Moreover, the fraction of neurons that exclusively encode prediction errors across stimulus dimensions decreases as stimulus dimensionality increases (Fig. 4d, right). Thus, neurons with mixed representations of stimuli and prediction-errors are prevalent in the network, while dedicated prediction-error neurons become increasingly rare, especially in high-dimensional scenarios.

## Experimental evidence for desegregated predictive representations

We then turned to testing this key prediction of our network model, by looking for signatures of mixed representations of predictions and stimuli in experimental data. In our recent work, we recorded primary auditory cortex responses in mice that were trained to associate a simple behavior, pressing a lever, with a simple outcome, a predictable tone[13]. Following extensive training, we made extracellular recordings from auditory cortex while animals were presented with probe auditory stimuli that differed from the expected stimulus along a variety of different dimensions, and while animals either pressed the lever or heard the tone passively (Fig. 4e).

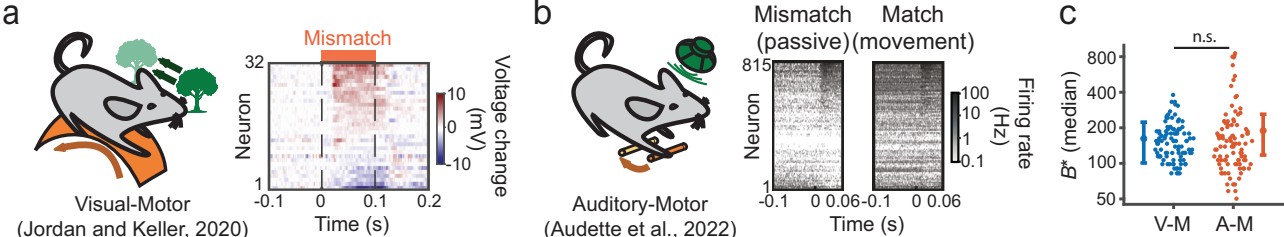

**Fig. 3 | Estimating the balance level from predictive coding experiments.**
**a** Schematic of a learned visual-motor association between running and virtual reality visual flow[20]. Voltage levels of different neurons ($n$ = 32 neurons) in primary visual cortex reveal tuning to mismatch between running speed and visual flow (prediction-errors). **b** Schematic of a learned audio-motor association between a lever press and a sound[12]. Neurons' firing-rates reveal tuning to auditory stimuli presented without (passive, prediction-errors) and with a lever press (movement).

Data from $n$ = 8 animals. **c** Estimating the median optimal balance level for V-M (blue) and A-M (red) experiments gives similar values. We assume that $\alpha$ = 0 based on the fact that the animals underwent extensive training on a single pair of stimuli in both experiments. Error bars are based on subsampling ($n$ = 100 repeats, Methods). Two-sided, unpaired $t$-tests are used ($p$ = 0.12). Circles and error bars: mean ± 1 S.D.

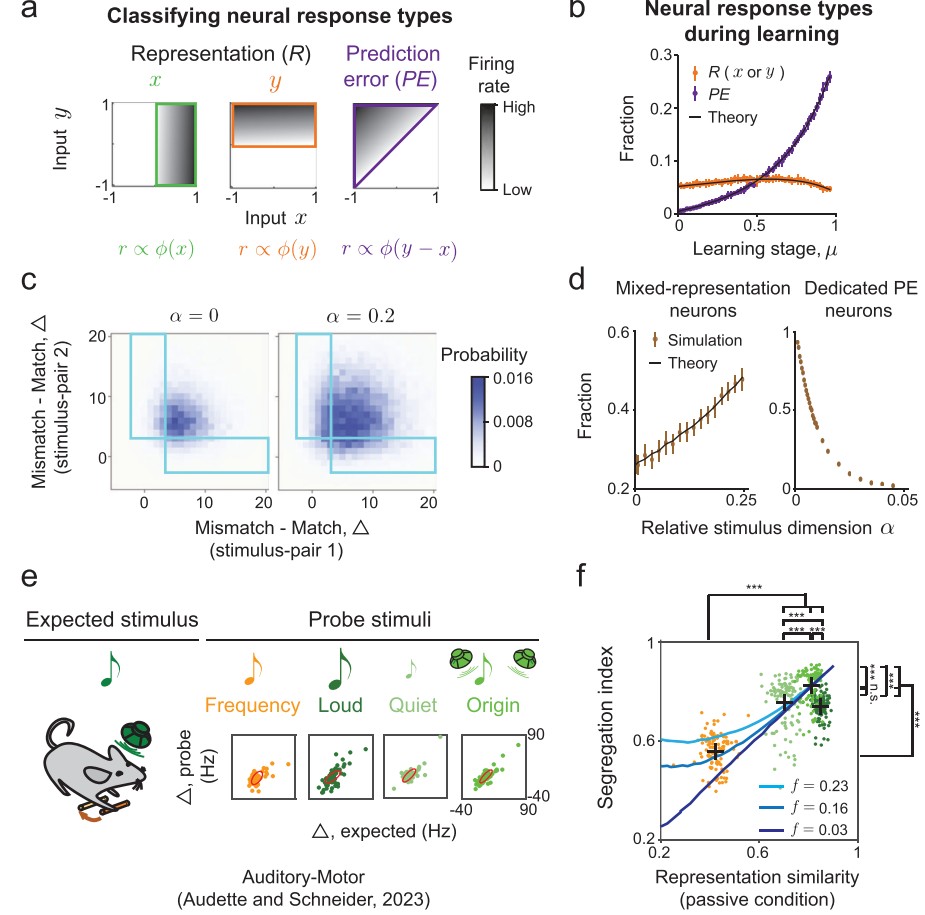

**Fig. 4 | Desegregated stimulus and error representations in networks performing high-dimensional predictive processing. a** Schematic of typical tuning profiles of different functional cell-types to the stimuli $x$ and $y$. **b** Fraction of representation ($R$) and prediction-error ($PE$) neurons in the model at different learning stages. Error bars: 1 S.D. computed over $n$ = 10 instances of the network. **c** Joint distribution of individual neurons' $\Delta$ values, the difference between mismatch and match responses to two specific stimulus-pairs. Only neurons responsive to both stimulus-pairs are included in the distribution (Methods). Mixed representation neurons have significantly different $\Delta$ values for the two stimulus-pairs, i.e., they are in the blue rectangular regions. **d** Effects of stimulus dimensionality on mixed representations. The fraction of mixed representation neurons increases as stimulus dimension increases (left). The fraction of dedicated prediction error (PE) neurons decreases as the stimulus dimension increases. See SI §6 for

definition of dedicated $PE$ neurons. Circles: mean; Error-bars: 1 S.D. computed over $n$ = 200 instances of the network. **e** Evaluating the segregation of stimulus and prediction-error representations based on neural recordings during a learned auditory-motor association. Shown are the $\Delta$ values of stimulus-responsive neurons for the expected sound and each probe type (colors). The length and direction of major and minor axes of the red ellipses correspond to the amplitude and direction of the two leading principal components of the data ($n$ = 5 animals). **f** Segregation index versus representation similarity for different pairs of expected and probe sounds. Colored points correspond to subsamples of the data, and crosses correspond to the average for each probe type (Methods). Experimental data is compared with equivalent quantities from the model, by varying the sparsity of responses in the model ($f$, see SI §4).

Here we analyzed this data as follows. For each neuron, we computed the difference ($\Delta$) between the mismatch (passive: sound only) and match (active: lever press + sound) neural responses (Fig. 4e, bottom), similar to our analysis of the neural activity in the model (Fig. 4c). Note that for each of the four probe sounds, "match" corresponds to a lever press paired with the probe sound, while "mismatch" corresponds to responses following the probe sound without a lever press. We expected $\Delta$ values of mixed neurons to lie in the upper left or lower right corners of the plot (similarly to Fig. 4c, blue rectangles). This would correspond to neurons with match and mismatch responses that are similar for the expected sound but differ for the probe sound, or vice versa.

We quantified the degree of mixing, or desegregation of the predictive representation, by computing the Pearson correlation coefficient of the $\Delta$ values corresponding to the expected sound and each probe sound separately (Fig. 4e). We defined this coefficient as the segregation index, which is close to 1 if the $\Delta$'s are strongly correlated between the two stimulus-pairs (expected, probe). A segregation index close to 0 means that the representations of stimuli and predictions are "maximally mixed". We additionally computed representation similarity between the expected and probe sounds, as the correlation between neural responses to those stimuli. Crucially, representation similarity was based on neural responses in a separate experimental window during which sounds were presented passively, not following a lever press[13]. If neurons are segregated into two functional classes, the segregation index should be close to 1, irrespective of the representation similarity. By contrast, we found that the segregation index depends strongly on the representation similarity (Fig. 4f). Specifically, when the expected and probe sounds are similar (Fig. 4e, f, green shades), the segregation index is close to 1, although a random subsampling analysis indicates a statistically significant effect of the representation similarity on the segregation index. When the probe differs from the expected sound more substantially (Fig. 4e, f, orange), the segregation index exhibits a marked drop to ~ 0.5 ($P < 0.0005$, two-sided unpaired $t$-test; Methods). The relation between representation similarity and degree of segregation that we found is consistent with the prediction of the network model with mixed representation, with an appropriate level of coding sparsity (Fig. 4f). The significant dependence of the segregation index on the representation similarity and the fact that the segregation index is substantially smaller than 1, suggest that predictive processing is mixed in the mouse auditory cortex. A similar relationship was found when we used the "complementary" mismatch response to compute the $\Delta$'s, i.e., based on the neural response to a lever press with no sound, rather than a sound with no lever press (Fig. S5).

We note that the analysis presented here is an indirect test of the model prediction that predictive representations are mixed. Indeed, the desegregation in the model involves two learned stimulus-pairs (Fig. 4c), while in the experiment the animal was only trained on the expected sound. Nevertheless, the decreased segregation index we found for probe sounds markedly different from the expected sound provides strong evidence against the notion that the predictive processing circuit is functionally segregated into separate neural populations. The modeling approach we adopted here provides a framework for generating hypotheses that can be tested more directly in future experiments.

## Multimodal high-dimensional predictions of transient stimuli

Previously, we assumed that the stimulus inputs change slowly with respect to the neural dynamics, and leveraged this assumption to analytically compute properties of steady-state predictive representations. Here, we investigate the temporal dynamics of the predictive representations in a network presented with fast-varying stimuli. We analyzed the network responses to pulses of stimulus-pairs

under three conditions: $x$-only, $y$-only, and match (Fig. 5a). Crucially, the $x$ and $y$ pulses in the match condition are separated by an interval $\Delta t$, so the network does not receive those multimodal inputs at the same time, and must generate and maintain the prediction of $y$ after the stimulus $x$ is removed. This scenario is similar to our experiments, where the predictable sound appears a certain time after the lever-press[12]. We found that after learning, a slower timescale emerges in the network responses (Fig. 5a), which does not exist before learning, and could support prolonged yet transient predictive representations. Indeed, within this slow timescale, the average firing rate is suppressed in the match condition compared to the mismatch conditions ($x$-only, $y$-only), especially for small $\Delta t$ (Fig. 5a). This suppression is stimulus-specific: it is absent when $x$ and $y$ stimuli belonging to different pairs are presented (compare purple solid and dashed curves in Fig. 5a). We further found that suppression of network responses in the match condition is enhanced during learning, and depends strongly on the temporal association between the paired stimuli (Fig. 5b). These properties of the network model are consistent with our measurements of motor-auditory associations[12], in particular with our experiments where the timing of motor-auditory associations is perturbed[13].

We sought to further understand the transient predictive representations beyond the average firing rate of the network. We computed a time- and stimulus-dependent readout axis $\Delta r(t)$, defined as the difference between the firing rate vectors in the two stimulus conditions (Fig. 5c). Similarly to the average firing rates, after learning, the magnitude of $\Delta r(t)$ decays slowly, corresponding to preservation of stimulus-specific predictions within the network. To investigate how stimulus dimensionality influences the functional cell-type-identities of neurons in the network, we computed a time-dependent joint distribution of $\Delta$'s, the difference in responses between the mismatch and match conditions, for two stimulus-pairs (Fig. 5d). When applying the functional cell-type classification criteria to this distribution, we found that the number of mixed-representation neurons in the network changes over time. Similarly to our analysis of the steady-state responses, the number of mixed representation neurons increases as the stimulus dimensionality increases (Fig. 5e).

We also probed our network model with finite-width step inputs (Fig. S6). Remarkably, the readout directions to decode the stimulus condition and the categorization into functional cell-types that were computed based on steady-state responses also apply here. Neurons with $R$ or $PE$ responses to a particular stimulus-pair at steady-state typically belong to the same functional class during the transients. An important advantage of this type of time-dependent stimuli is that it allowed us to investigate the temporal dynamics of the balance level in different stimulus conditions, which cannot be defined in the case of pulse stimuli due to the absence of feed-forward input. After learning, the network exhibits a transient tight balance after the onset of the second (predicted) stimulus (Fig. S6c). The peak of the balance level during this period decreases as stimulus dimension increases (Fig. S6d), similar to the steady state predictive representation (Fig. 2f). Taken together, our results demonstrate that the mechanisms underlying predictive computations and the properties of high-dimensional predictive representations identified based on steady-state responses also apply in scenarios where stimuli are presented transiently.

We tested the robustness of the computations performed by our network model to random pruning of synaptic connections, which leads to asymmetric connectivity (Fig. S7). The sparse network exhibits sustained activity fluctuations even when the stimulus inputs are constant, consistent with a previous study[63]. Despite those strong fluctuations in individual neurons' activity, the stimulus-specific internal prediction generated by the network remains stable (Fig. S7). Increasing the stimulus dimensionality in the sparse network model leads to a more loose balance and a larger fraction of mixed-representation neurons (Fig. S7).

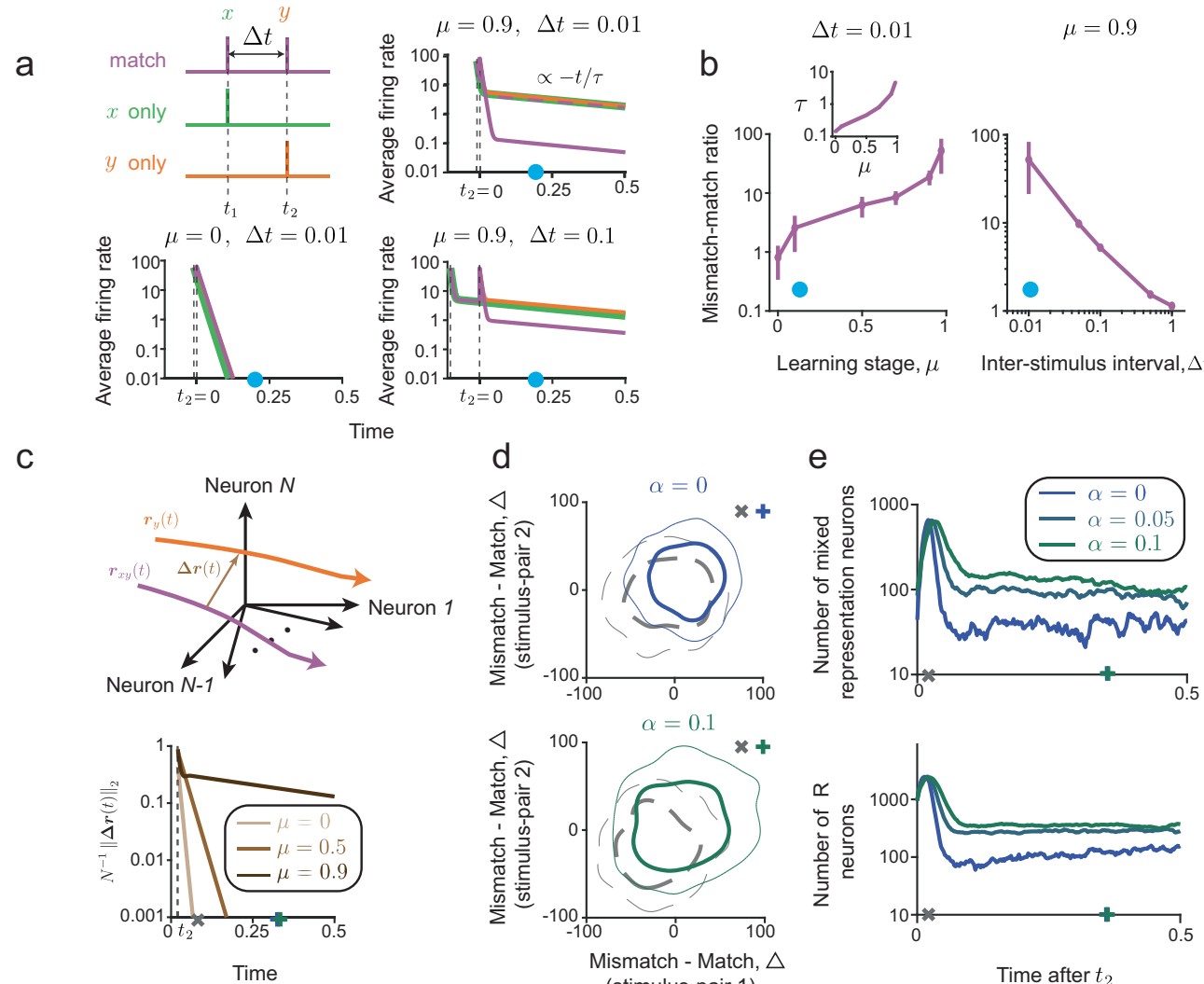

**Fig. 5 | Transient neural dynamics in predictive processing networks. a** Average firing rates in response to mismatched pulse stimuli decay rapidly before learning and slowly after learning ($\mu = 0$, $0.9$, respectively). The response in the match condition is suppressed when the inter-stimulus interval is short ($\Delta t = 0.01$). Response to matched $x$ and $y$ stimuli that belong to different pairs (dashed) overlaps with mismatched responses. **b** Ratio of average mismatch and match responses as a function of learning stage and inter-stimulus interval, computed after stimulus removal (blue circles in (**a**)). Inset: The decay timescale of neural responses increases during learning. Line: mean; Error-bars: 1 S.D. computed over $n = 10$ instances of the network. **c** Illustration of transient neural trajectories elicited by stimuli in the mismatch and match conditions. After learning, the distance

between the trajectories remains large even after the stimuli are removed, suggesting that learning enhances the discriminability of the two stimulus conditions. **d** Time-dependent joint distribution of individual neurons' $\Delta$ values: differences between mismatch and match responses to two specific stimulus pairs. Shown are distributions at two time points corresponding to gray, blue, and green symbols in (**c**)). Thick and thin lines respectively enclose 50%, 80% of the distribution. Only neurons responsive to both stimulus-pairs are included in the distribution (Methods). **e** Time-dependent numbers of mixed- and $x$-representation neurons for different stimulus dimensionalities $\alpha$. As the stimulus dimension increases, the total number of mixed-representation neurons increases and remains large for a longer period.

## Predictive processing in excitatory–inhibitory networks

Thus far, we have focused on relating neural responses in the model to measurements of excitatory neurons' activity[12,13,16]. Each neuron's projections in our network could be both excitatory (E) and inhibitory (I), so it does not obey Dale's law. Given the growing literature on the role of inhibitory neurons in computing predictions[44,45], we sought to link the modeling results to experiments more tightly by extending it to a network with separate E and I neurons. We did so by requiring that the activity of E neurons in the E/I network matched exactly that of neurons in the original model. This guarantees that the E neurons possess the predictive coding properties we studied so far, and opens the door to study the functional role of I neurons. The connectivity in the E/I network has four components, corresponding to synapses to and from E and I neurons (Fig. 6). We used non-negative matrix factorization to "solve" for those components (Methods, refs. 64,65). The

balance level $B$ defined previously based on feedforward and recurrent inputs (Fig. 2), is equal to the stimulus-specific component of the E/I balance in the E/I networks (SI §5).

The aforementioned mathematical procedure did not yield a unique connectivity structure. Rather, we found a one-parameter family of connectivity structures that all meet those constraints. This parameter, denoted $\lambda_{EI}$, interpolates between two extremes of structured E/I connectivity (Fig. 6b). In one extreme ($\lambda_{EI} = 0$), inhibition is "private": Each "parent" E neuron projects to a single "daughter" I neuron with equal activity. This has been an implicit assumption of previous predictive coding models with lateral inhibition[32,47]. In the opposite extreme ($\lambda_{EI} = 1$), each I neuron receives a large number of excitatory inputs and signals an "internal prediction" of one stimulus learned by the network, similar to previous models with segregated neural populations[44,45]. We investigated the continuum of inhibitory

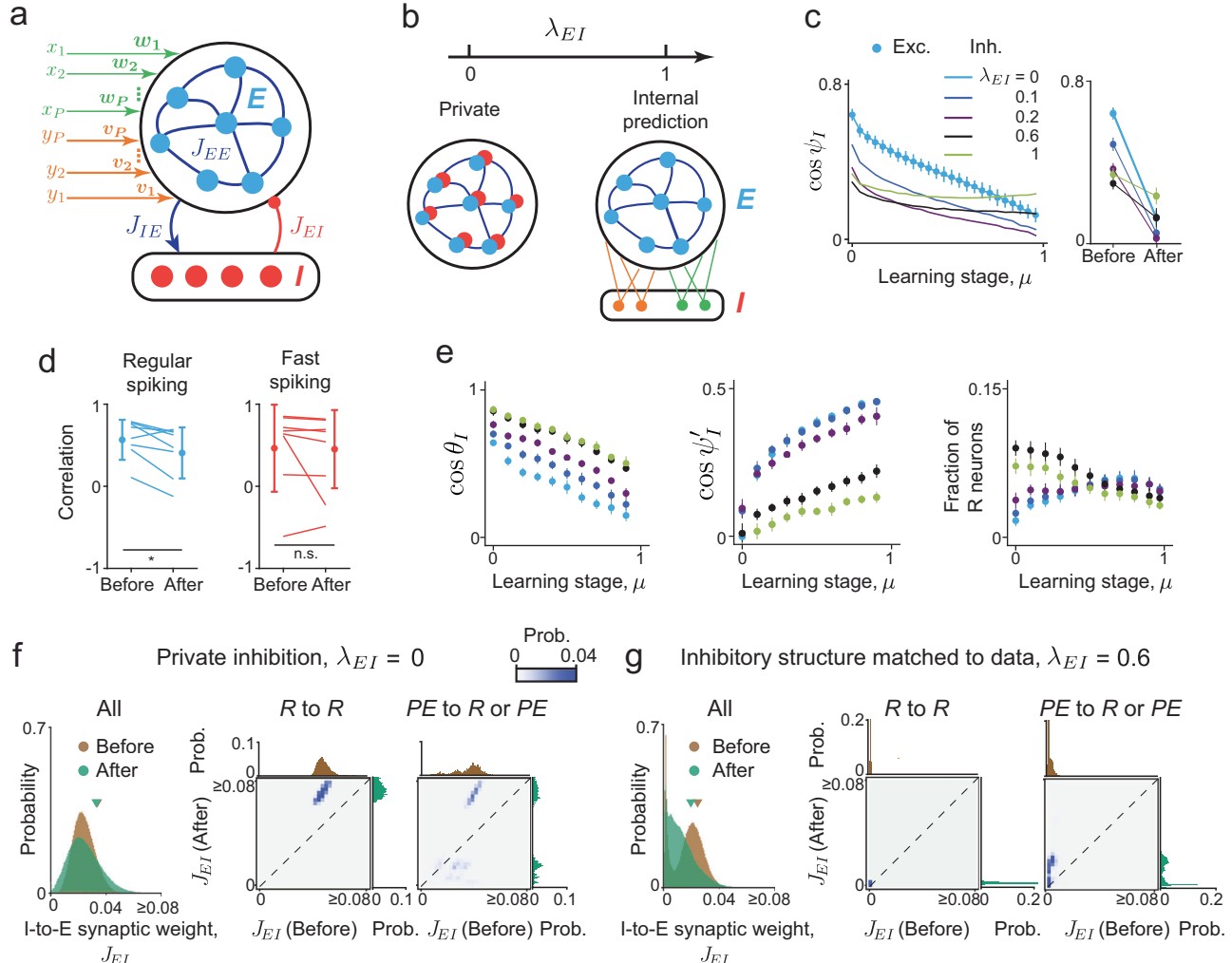

**Fig. 6 | A data-constrained excitatory/inhibitory model suggests that internally generated predictions are distributed across the network. a** Schematic of the E/I network. Excitatory neurons receive external inputs, and their activity is constrained to equal that of neurons in our original model. **b** A family of E/I network solutions is parameterized by $\lambda_{EI}$. **c** The cosine similarity ($\cos \psi_I$) between the match and mismatch inhibitory responses to stimulus $x$ ($\mathbf{r}_{xy}$, $\mathbf{r}_x$), for different values of $\mu$ and $\lambda_{EI}$ (left). Error bars: 1 S.D. computed over $n = 200$ instances. **d** Analogous correlation between population responses, computed separately for regular-spiking (RS) and fast-spiking (FS) neurons from $n = 8$ animals in ref. 12. Each point represents data from one animal. Mean ± 1 S.D. across animals is also shown. RS neurons significantly decorrelate during learning ($p = 0.025$), while FS neurons'

correlation does not change ($p = 0.31$, two-sided unpaired $t$-test). **e** The angle $\theta_I$ (left) between inhibitory population responses to the paired stimuli in the mismatch conditions ($\mathbf{r}_x$, $-\mathbf{r}_y$), the angle $\psi_I'$ (center) between match and mismatch inhibitory population responses to stimulus $y$ ($\mathbf{r}_{xy}$, $\mathbf{r}_y$) and the fraction of inhibitory $R$ neurons (right). Error-bars: 1 S.D. computed over $n = 20$ instances. **f** Synaptic weight distribution of all I-to-E connections before and after learning, when $\lambda_{EI} = 0$ (left), and for pairs of E and I neurons belonging to specific functional classes ($R$ to $R$, middle; $PE$ to $R$ or $PE$, right). **g** Same as (**f**), when inhibitory structure is matched to data ($\lambda_{EI} = 0.6$). Here, learning sparsifies and depresses inhibitory connections. Connections between $R$ neurons remain very small throughout learning.

representations between these extremes using the same approach applied to E neurons (Figs. 1b–e and 4b). We started with the alignment of inhibitory responses to stimulus $x$ in the match ($\mathbf{r}_{xy}$) and mismatch ($\mathbf{r}_x$) conditions, at different learning stages (Fig. 6c). Before learning ($\mu = 0$), increasing $\lambda_{EI}$ leads to a marked decrease in the alignment of inhibitory responses. After learning ($\mu \approx 1$), increasing $\lambda_{EI}$ leads to a non-monotonic effect on alignment. Intriguingly, for $\lambda_{EI} = 1$, after learning, the alignment of I responses in the two conditions is larger than that of E responses (Fig. 6c, compare green and black for $\mu = 1$).

These properties allowed us to estimate the parameter $\lambda_{EI}$ based on empirical measurements of regular-spiking (RS, putative excitatory) and fast-spiking (FS, putative inhibitory) neurons. To achieve that, we computed the correlation between auditory cortex match and mismatch responses, separately for RS and FS neurons recorded in ref. 12, and then compared those correlations to the model before and after learning (Fig. 6d). Specifically, the pairing between movement

and a probe sound (not presented during training) was regarded as before learning and the pairing between movement and the expected sound as after-learning (Methods). This correlation decreased significantly during learning for RS neurons, consistent with the change in the model's E population responses (Fig. 6c, blue circles). By contrast, correlation of FS population responses did not change significantly during learning, which rules out small values of $\lambda_{EI}$. Moreover, the correlation value after learning was similar for RS and FS neurons, which rules out large values of $\lambda_{EI}$. Taken together, our analysis suggests that an intermediate value of $\lambda_{EI} \approx 0.6$ best captures the experimental observations, consistent with the suggestion of "promiscuous" inhibitory connections mediating suppression of expected stimuli[11].

Given this experimentally-constrained value ($\lambda_{EI} = 0.6$), our theory generates testable predictions for inhibitory predictive representations. First, we expect that anti-alignment of mismatch I responses ($x$-only, $y$-only) is significantly weaker when compared to anti-alignment

of E responses in the same conditions (Fig. 6e, left; Fig. 1d, e). Second, we predict large correlations between inhibitory responses in the match and $y$-only mismatch conditions (Fig. 6e, middle), when compared with E responses. The asymmetry of $r_x \cdot r_{xy}$ and $r_y \cdot r_{xy}$ overlaps in the model may in the future, be related to distinct functional responses of inhibitory neuron subtypes[23,66]. Third, the fraction of I neurons with $R$ responses decreases moderately during learning, compared to E neurons. We note, however, that the fraction of E neurons with $R$ responses shows moderate dependence on the threshold, particularly before learning (Fig. S8), which may make it challenging to detect differences in fractions of neurons with $R$ responses between E and I neurons.

Previous work on predictive coding suggested that associative learning enhances top-down inhibitory projections from outside the local circuit[2,16], which cancel bottom-up excitation and suppress neural responses in the match condition. We therefore wondered what changes in inhibitory connectivity during learning lead to stimulus-specific suppression of neural activity in our E/I network model. One option is that inhibitory connections that predict the stimulus are strengthened[2]. Alternatively, inhibition could undergo more subtle reorganization such that inhibitory signals are distributed differently before and after learning.

We calculated the distribution of I-to-E synaptic weights before and after learning in the family of E/I network models. When inhibition is private ($\lambda_{EI} = 0$), this distribution broadens during learning (Fig. 6f). Examining the change in synaptic weights conditioned on the functional cell-type of pre- and post-synaptic neurons ($R$ or $PE$), suggests that stimulus-specific suppression of E responses arises from potentiated I synapses from neurons "faithfully" representing the stimulus. In other words, when inhibition is private, the predictive signal arises in part due to strengthened projections from inhibitory $R$ neurons to excitatory neurons (Fig. S9). By contrast, when inhibitory structure was matched to experimental data ($\lambda_{EI} = 0.6$), learning leads to overall sparsification of I connections (Fig. 6g). Interestingly, here $R$-to-$R$ connections can be either potentiated or depressed, unlike the $\lambda_{EI} = 0$ case (compare middle panel of Fig. 6f, g). Moreover, when $\lambda_{EI} = 0.6$, inhibitory connections originating from $PE$ neurons that are initially very weak get strongly potentiated.

Together, our results suggest that (*i*) Predictive processing is learned without large increases of the average inhibitory connection strength. This was also seen for other values of $\lambda_{EI}$ (Fig. S10). (*ii*) The "strategy" for learning predictive processing can differ substantially, and depends on the underlying circuit structure (different values of $\lambda_{EI}$ in the model). (*iii*) When inhibitory structure is matched to data, the "internal model" is highly distributed and, surprisingly, arises in part from potentiated connections from inhibitory neurons signaling prediction-error. Another signature of this distributed strategy is the decrease of total inhibitory input to each excitatory neuron during learning (Fig. S10), which suggests that predictions are primarily computed by recurrent circuitry rather than directly from top-down inputs.

### Predictive representations in hierarchical neural networks

Sensory brain regions are known to have a laminar structure and distinct layer-specific response characteristics in associative learning tasks[17,20,67]. In the context of the task involving sensorimotor predictions, it has been suggested that motor-related input originates from motor regions and first enters the primary sensory region via deep layers (L5/6)[2,50,68,69]. On the other hand, the bottom-up sensory-related inputs first enter the primary sensory region via L4, which further projects to L2/3, where the bottom-up and top-down inputs are integrated and processed[70,71]. To investigate the effects of the laminar structure on predictive processing, we extended the recurrent network model, which has a single-module and no hierarchical structure, to a network model with three recurrently interconnected modules

(Fig. 7). During associative learning, the network receives paired multimodal inputs. Crucially, the first module (M1) of the network receives inputs from one modality, and the last module (M3) receives inputs from the other modality (Fig. 7a). Differently from previous studies[1,17,37,39], each module in this network computes bidirectional predictions, corresponding to inputs from the level above and below it in the hierarchy. For example, M2 computes predictions of activity in M1, simultaneously with predictions of activity in M2 computed in M1. Thus, this hierarchical model can also be applied to cross-modal processing performed by distinct brain regions that exchange predictive signals bidirectionally (e.g., auditory and visual cortices, ref. 16), beyond laminar organization within a single brain region.

We first studied the effects of module-specific gain parameters. After learning, the average mismatch responses increase monotonically with $b_1$ and $b_2$ (Fig. 7b). We constrained the average mismatch response to be larger than a certain threshold value and minimized the match responses for each module. Doing so gave a continuous set of parameter combinations for which the network satisfies those constraints (Fig. 7b, magenta line). We fixed $b_2$ such that the fraction of prediction error neurons in M2 after learning is similar to the fraction in the single-module model (Fig. 3b), which also fixes $b_1$ and $b_3$ (Fig. 7b, star). With these constrained parameters, we assessed how associative learning shapes neural representations across different modules.

In the $x$-only mismatch condition ($x = 1$, $y = 0$), the overall mismatch responses increase during learning, with notable module-specific differences (Fig. 7c): neurons in M1 that directly receive the $x$-stimulus input have remarkably similar responses in the match and mismatch conditions throughout learning. In contrast, neurons in M3 respond predominantly to stimulus $y$ but gradually become tuned to stimulus $x$ as learning progresses. Neurons in M2 exhibit the largest mismatch-match response ratio and develop the most significant prediction error responses after learning. To illustrate the role of bidirectional predictive processing, we compared cross-modal responses in our hierarchical model with a model that assumes unidirectional predictions[1,37] (Fig. S11). Specifically, in the unidirectional model, M2 generates predictions of activity in M1 and M3, but no "backward" predictions are generated from M1 or M3 to M2. Minimizing such one-way prediction errors implies a different objective function optimized by the network, and results in architecture and neural dynamics that are different from those given by a model with bidirectional predictions (Fig. S11a). We set the parameters of the two models to be equal to facilitate comparisons, and found that neurons in M1 and M2 of both models exhibit similar response profiles in the match and $x$-only mismatch conditions. However, responses of neurons in M3 in the unidirectional model are weakly modulated by the (cross-modal) stimulus $x$ (Fig. S11b). By contrast, in the bidirectional model, M3 develops robust cross-modal responses (Fig. 7c). Consequently, decoding analyses confirm that M3 activity carries substantially less information about the sensory input to M1 in the unidirectional model compared to the bidirectional model we propose, across various stimulus conditions (Fig. S11c). Thus, computation of bidirectional predictions enhances cross-modal interactions between brain areas, consistent with recent experimental observations that motor areas encode auditory information[72] and that predictive signals are exchanged reciprocally between visual and auditory areas[16].

Next, we categorized neurons along the hierarchy into functional cell-types. Before learning, neurons activated by the stimulus $x$ independently of $y$ (i.e., $x$ representation neurons) are concentrated in M1–the module receiving the stimulus $x$ input directly. During learning, $x$ representation neurons also arise in M2 and M3, though the overall fraction of these neurons decreases from M1 to M3 (Fig. 7d). $PE$ neurons are initially very rare and emerge in all modules during learning, with the largest fraction concentrated in M2 (Fig. 7e). These results are consistent with the activity of layer-specific primary sensory cortex neurons[12,20,68].

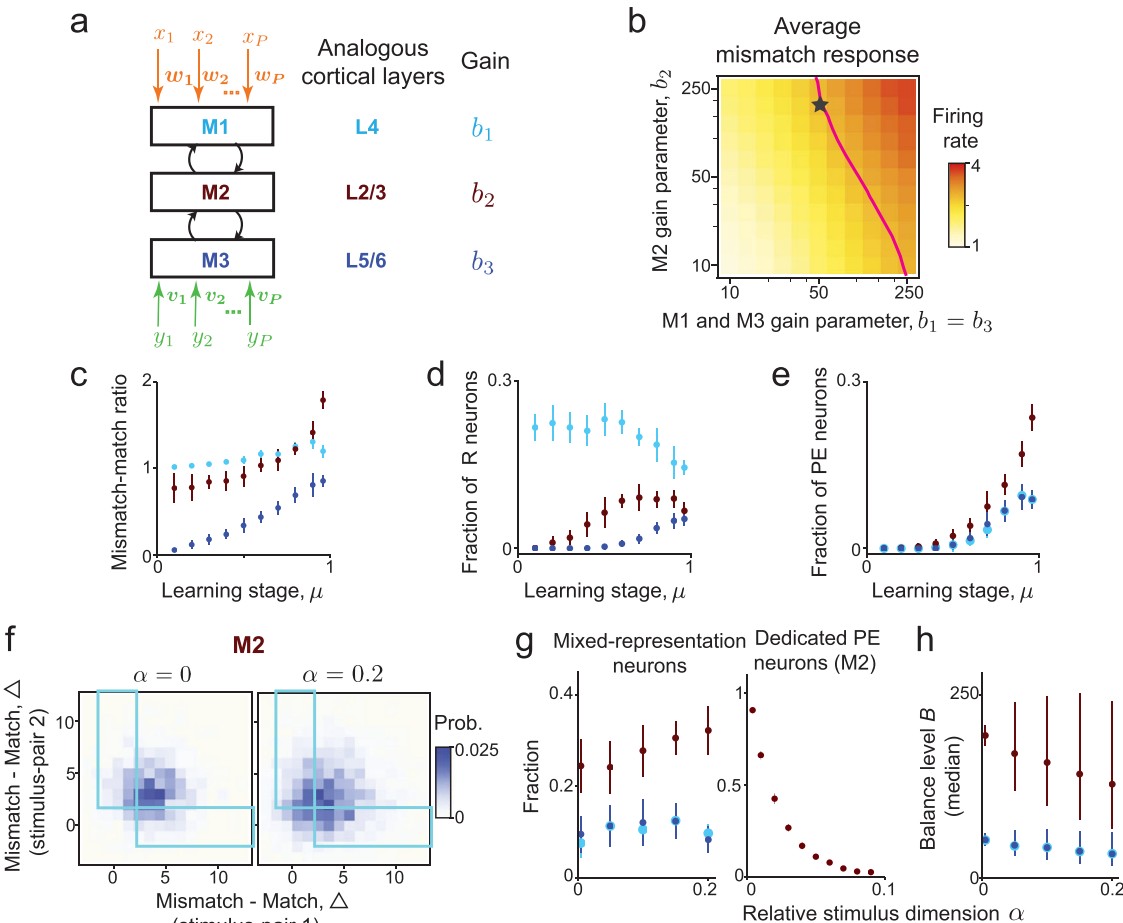

**Fig. 7 | Representations of stimuli and prediction errors vary across a hierarchical network. a** Hierarchical network for predictive processing with three modules. M1 and M3 receive stimulus $x$ and $y$ input, respectively. **b** The average $x$-only mismatch response increases with the module-specific gain parameters $b_{1,2,3}$. Line: mismatch response amplitude used to constrain $b_{1,2,3}$. Star: parameter values further constrained based on the fraction of prediction error neurons in M2, used in (**c**–**h**). **c** The ratio between the average firing-rates in the $x$-only mismatch and match conditions increases during learning. The increase is most prominent in M2. **d** The fraction of $x$ representation (R) neurons at different learning stages. Differences between the modules diminish with $\mu$. **e** The fraction of prediction error (*PE*) neurons at different learning stages. **f** Joint distribution of individual neurons' $\Delta$

values, defined as the difference between mismatch and match responses to two specific stimulus-pairs in M2. Mixed representation neurons are in the blue rectangular regions. The fraction mixed representation neurons increases with the stimulus dimension $\alpha$. **g**, **h** Effects of increasing the stimulus dimension $\alpha$. **g** The fraction of mixed representation neurons increases with $\alpha$ in M2, and is unchanged in M1 and M3 (left). The fraction of dedicated prediction-error (PE) neurons in M2 decreases as the stimulus dimension increases (right). **h** The median balance level decreases with $\alpha$ in M2 and remains approximately constant in M1 and M3. Circles and error-bars: mean ± 1 S.D. **c**–**e**, **g**, median and IQR (**h**). Results are based on $n = 30$ instances of the network.

We finally evaluated the network responses for two stimulus-pairs. Similar to the single-module network model, mixed representation neurons arise in all modules after learning (Fig. 7f, g). The fraction of mixed representation neurons is maximal in M2, and it increases with the number of learned stimulus-pairs (Fig. 7f, g). Similar to the single-module network (Fig. 4d, right), the fraction of dedicated error neurons in M2 decreases rapidly as stimulus dimensionality increases (Fig. 7g, right). We also found that the more pronounced desegregation of neural representations is accompanied by a significant decrease in the median balance level in that module (Fig. 7h), suggesting that loose balance is the underlying circuit mechanism supporting the mixed predictive responses at the cellular level. Unlike our findings in M2, the fraction of mixed representation neurons and the median balance level in M1 and M3 do not show strong dependence on the stimulus dimensionality. These results highlight the impact of anatomical structure on shaping network function. Specifically, we found that different modules have different fractions of representation and prediction error neurons, reminiscent of recent experimental findings[18]. However, despite this heterogeneity, representations of

stimuli and prediction error are desegregated in all modules after learning.

## Discussion

Here, to understand how the brain constructs an internal world model that guides behavior in natural conditions, we investigated the neural representations formed in a class of recurrent neural networks that learn to generate multimodal and high-dimensional predictions. Our mathematical analysis reveals key neural mechanisms supporting high-dimensional predictive coding, generates novel testable hypotheses for functional properties of the corresponding neural circuits, and provides a framework within which experimental data of large-scale neural recordings can be quantitatively analyzed. These results extend previous studies on neural representations of unimodal sensory signals[1,33,40] and offer insights on how multimodal responses are organized within primary sensory areas, as observed in recent experiments[2,16,50]. Additionally, the approach we adopted here allows incorporation of information on cell-types and anatomical structure into the model, which can elucidate their roles in predictive computations.

We focused on a recurrent network model (Fig. 1) for two reasons. First, cortical circuitry that performs predictive processing is known to be highly recurrent. Plasticity of recurrent connections forms functional neuronal assemblies[73], which were suggested to underlie behaviorally-relevant sensory discrimination[74]. Second, predictions for sensory stimuli typically unfold over time, which can be naturally implemented by intrinsic dynamics of recurrent networks[40,75]. The recurrent network structure we derived here performs predictive computations at steady-state, for mathematical tractability. We demonstrated that our network model also performs predictive computations when probed with time-dependent stimuli. Importantly, the mechanisms supporting those computations generalize from the steady-state to the transient case (Fig. 5). An interesting direction for future research is to extend our results for a network with connectivity "prescription" to networks that can learn predictions online (continually)[76,77].

Our results suggest that a balance between feedforward and recurrent input, or indeed between excitation and inhibition, can lead to robust internal predictions within local circuits. This has been suggested previously in predictive coding modeling studies[40,41,43,78,79]. An important novel prediction revealed by our analysis is that, when extending these models to realistic conditions (with multimodal high-dimensional inputs), there is an optimal, finite balance level, which decreases with stimulus dimension (Fig. 2). Our theory further suggests that a network with infinitely high balance[41] could be especially vulnerable to noise in high-dimensional scenarios.

Based on our results, we hypothesize that the large degree of heterogeneity of empirical E/I balance levels in different experiments[61] may be a signature of the differences in the stimulus ensembles animals were exposed to. Our results in Figs. 2 and 3 suggest that this hypothesis could be tested systematically by exposing animals to increasingly rich sensory environments. Here, too the temporal dynamics of the network may be important, as synaptic delays may affect the optimal degree of balance in circuits performing low-dimensional predictions[43,80].

The role that balance plays in computing predictions has important implications for the source of predictive signals and the timescale of learning them. (i) Previous work has shown that cross-modal predictions are often stimulus-specific[12,16,60]: signals from one brain region can suppress responses to a particular predictable stimulus in another region (e.g., motor cortex activity suppressing visual cortical responses). It is notable that within the model we studied here, those computations are performed without fine-tuning long-range projections[2]. Rather, local recurrent connections in the "receiving region" can extract the predictions from long-range inputs with "promiscuous" connectivity[11], relying on E/I balance and activity-dependent synaptic plasticity. (ii) Prediction-error responses in the same cortical region can arise at very different timescales, from as little as minutes[26] to days of training[12,16]. We believe that the diversity of the identified E/I balance mechanisms (e.g., firing-rate adaptation, synaptic-scaling, Hebbian plasticity; see review in ref. 53), may explain this wide temporal range of predictive processing learning dynamics. Future work may reveal that the model we studied here has explanatory power also for the emergence of predictions over faster timescales than the experiments considered here, and thus could be applied to predictive processing circuits in subcortical regions and in invertebrates.

An important finding of our work is that predictive representations are desegregated: neurons that signal prediction-errors for one stimulus-pair may faithfully represent the presence of stimuli for a second pair. Based on experiments where animals were probed with multiple types of unexpected sounds, we found signatures of this desegregation at the cellular level in mouse auditory cortex (Fig. 4). Specifically, the segregation index computed from the data deviates significantly from the hypothesis of functionally segregated neural populations, supporting instead a mixed representation of stimulus and error signals in layer 2/3 of the primary auditory cortex. Our current work qualitatively accounts for the desegregation seen in the data based on a mechanistic circuit modeling approach (Fig. 4f). Nevertheless, the data analysis performed here does not rule out the possibility that dedicated prediction-error neurons may exist in other cell-types that were not recorded in our dataset. Future experiments targeting a broader diversity of cell-types will be necessary to test this possibility. Another recent study in mice performing multiple stereotyped motor actions reported mixed representations of the motor variables and reward prediction-errors across the neocortex[81], as suggested by the network model for high-dimensional scenarios. An interesting future direction would be to investigate potential functional advantages of such mixing of motor variables, sensory predictions, and reward predictions, and their quantitative dependence on the task structure from a normative approach.

Previous work often explicitly assumed that separate neural populations encode prediction and prediction errors[17,37–39]. Here, we adopted a different strategy, similar to classical sparse coding models[33–35,46], that does not impose this assumption. We found that the resulting recurrent network develops mixed neural representations as a direct consequence of minimizing the multimodal prediction errors under energy constraints. In this model, prediction error signals are computed in a distributed manner via recurrent connectivity, based on a circuit mechanism that is different from models with segregated neural populations.

Our findings are related to the expanding literature on mixed-selectivity[82–84], where neurons exhibit complex tuning to multiple stimulus features. While even a random network can exhibit mixed-selectivity[83], the neurons' tuning curves there are unstructured, which requires finely-tuned decoders to read out task-relevant variables. Here, we report neurons that have mixed-selectivity to internally generated predictions of sensory and motor variables (Figs. 4–6). Crucially, the learned neural representations in the model studied here are highly structured, and enable the reading out of different stimulus features without "re-learning" the decoder (Fig. 2).

Although neurons in the network model studied here and in electrophysiological recordings from the auditory cortex have mixed selectivity for stimuli and prediction-errors, the auditory cortex also contains neurons that more specifically encode prediction-errors[13]. Notably, the degrees of neurons' mixed selectivity to stimulus and error could also be layer-specific[12]. This is recapitulated by our hierarchical network model (Fig. 7). The hierarchical model also suggests that this layer-specific degree of mixed representation arises from the layer-specific changes of balance. This would be an interesting experimental prediction to test in future research. Recent work in the mouse visual cortex has identified specific genetic markers that are over-expressed in neurons encoding positive versus negative prediction errors[18]. However, differences in methodologies and the time course of analysis make direct comparisons across these studies challenging. As these experiments[18] involve learning a single association, it is difficult to distinguish between neurons with mixed representations and those that "purely" encode prediction errors. It remains possible that the sensory cortex contains a heterogeneous population of neurons—some that share roles in encoding both stimuli and prediction errors, and others that selectively encode one or the other. Future experimental studies in which animals are trained to learn multiple associations are needed to more directly resolve this question.

Our focus on a network responding to orthogonal ("one-hot" or "tabular") stimuli, motivated by choice of stimuli in recent experiments[2,9,11,12,16,20], allowed us to gain insight into the role of circuit structure in predictive processing, independently of potential effects of altering stimulus statistics. We demonstrated that our results extend

to scenarios with uncorrelated noise (Fig. S1). However, the simplified stimuli used here do not capture the complex correlation structure of high-dimensional natural stimuli. A comprehensive theoretical account of more general stimulus statistics that builds upon the analysis presented here is an important direction for future research, and would also require additional experiments—training animals to perform more complex tasks.

In summary, predictive processing is a ubiquitous and fundamental computation supporting diverse behaviors across animal species. Here, we take a first step towards bridging the gap between the theory of multimodal high-dimensional predictive processing and circuit-level neural recordings in predictive processing paradigms. Our results reveal the functional roles of specific circuit motifs and mechanisms in performing multimodal high-dimensional predictive processing. In a broader context, our work will advance the understanding of how the brain constructs complex internal-models by shedding light on commonalities and differences between biological predictive coding circuits and artificial systems, particularly those trained using self-supervised algorithms[39,85].

## Methods

### Recurrent network model

The network model consists of $N$ neurons whose firing-rates are described by the time-dependent vector $\boldsymbol{r}(t) = (r_1(t), ..., r_N(t))$. The network is driven by high-dimensional stimulus input, denoted $\boldsymbol{x}(t) = (x^1(t), ..., x^P(t))$ and $\boldsymbol{y}(t) = (y^1(t), ..., y^P(t))$. The vectors $\boldsymbol{x}$ and $\boldsymbol{y}$ correspond to stimuli from two modalities that are paired during training.

The dynamics of the recurrent network are given by

$$\frac{dh_i(t)}{dt} = -h_i(t) + b \left( \underbrace{\sum_{j=1}^{N} J_{ij} \phi(h_j(t))}_{-I_i^R} + I_i^F(\boldsymbol{x}(t), \boldsymbol{y}(t)) \right). \quad (1)$$

Here $h_i(t)$ is the voltage level of each neuron and is related to its firing-rate via a nonlinear activation function, $r_i(t) = \phi(h_i(t))$. Note that the input each neuron receives in Eq. (1) is decomposed into the recurrent ($I_i^R$) and feedforward ($I_i^F$) components. We rescaled the connectivity matrix $J_{ij}$ and the feedforward input $I_i^F(\boldsymbol{x}(t), \boldsymbol{y}(t))$ by a constant $b$, which can be interpreted as a gain parameter.

The explicit forms of $J_{ij}$ and $I_i^F(\boldsymbol{x}(t), \boldsymbol{y}(t))$ were determined based on a normative approach as follows (derivation details appear in SI §2). We assume that the neurons' dynamics jointly minimize the following objective

$$E(t) = \underbrace{\sum_{k=1}^{P} \left[ \left(x^k(t+d) - \hat{x}^k(t)\right)^2 + \left(y^k(t+d) - \hat{y}^k(t)\right)^2 \right]}_{\text{Prediction–errors}} + \underbrace{\frac{2}{b} \sum_{i=1}^{N} F(r_i(t))}_{\text{Encoding efficiency}}, \quad (2)$$

where $\hat{x}(t)$ and $\hat{y}(t)$ are the internal predictions generated by the network at time $t$ and $F(r)$ is a monotonically increasing function whose explicit form depends on $\phi$, the nonlinear activation function (SI §2.1). For ReLU nonlinearity $[\phi(z) = \max(z - \theta, 0)]$, $F(r) = (r + \theta)^2/2$. This is similar to previous work on predictive coding for natural images[34,35], where the response nonlinearity functions as a regularization term that controls encoding efficiency. Here, we focus specifically on network responses with multimodal inputs. Minimizing Eq. (2) is equivalent to performing Bayesian inference to extract the latent "cause" of the sensory signals (SI §2.2). This model is closely related to the classical Hopfield model for associative learning, but with anti-Hebbian learning rule[86]. We note that the parameter $b$ in Eq. (2) controls a trade-off between minimizing prediction-errors and maximizing encoding efficiency.

We further assume that the internal predictions are linear read-outs of the network activity

$$\hat{x}^k(t) = \frac{1}{N} \boldsymbol{w}^k \cdot \boldsymbol{r}(t), \quad \hat{y}^k(t) = \frac{1}{N} \boldsymbol{v}^k \cdot \boldsymbol{r}(t). \quad (3)$$

Here $\boldsymbol{w}^k, \boldsymbol{v}^k \in \mathbb{R}^N$ are the readout weight vectors. These internal predictions are, by definition, predictions of future input, as indicated by the delay $d$ in Eq. (2). However, we will focus on the scenario where the input changes much more slowly than the neurons' firing-rates. Therefore, on the timescale of firing-rate changes [Eq. (1)], we will regard the stimulus inputs to be approximately constant, i.e.,

$$x^k(t+d) \approx x^k(t) \approx x^k, \quad y^k(t+d) \approx y^k(t) \approx y^k. \quad (4)$$

Notice that Eq. (4) does not mean that the stimulus must be static, since a stimulus that varies on a slower timescale than neural activity satisfies this assumption. In that scenario, stimulus changes will be closely tracked by corresponding changes in neural activity. A particular instance of this scenario is the case with a static stimulus. In this case, the related predictive computation is commonly referred to as "spatial prediction" in the predictive processing literature[57]. Our results apply to both scenarios: slowly varying and static stimuli.

We assume that the weight vectors $\boldsymbol{w}^k$ and $\boldsymbol{v}^k$ change during learning so as to minimize the objective function $E(t)$ [Eq. (2)]. This optimization process can be viewed as weight-changes governed by a combination of gradient descent on the squared prediction error in Eq. (2), and homeostatic plasticity (SI §2.1). These weight changes can emerge from local plasticity rules applied to dendritic compartments (SI §2.3), extending the findings in refs. 46,47. If weights are initialized randomly, learning increases the correlation between the weight vectors (SI §2.1). Specifically, we show that in the large network size limit ($N \to \infty$), the weight vectors have the following statistics,

$$\begin{aligned} \langle w_i^k \rangle &= \langle v_i^k \rangle = 0, \\ \langle (w_i^k)^2 \rangle &= \langle (v_i^k)^2 \rangle = 1, \\ \langle w_i^k v_i^k \rangle &= \mu^k. \end{aligned} \quad (5)$$

Here, $w_i^k$ and $v_i^k$ are the components of $\boldsymbol{w}^k$ and $\boldsymbol{v}^k$, which have zero mean and unit variance due to homeostatic plasticity. The correlation between them is $\mu^k$, which increases during learning (i.e., as the objective function $E$ decreases). For simplicity, unless noted otherwise, all stimulus-pairs have the same "age", i.e., $\mu^k = \mu$ does not depend on the index $k$. We further assume that the weight vectors have multivariate Gaussian distribution. Under these assumptions, we obtained analytical solutions for the dependence of steady-state firing-rate distribution on the stimulus input and the correlation $\mu$ in two limits (SI §3): the high-dimensional case where both $N$ and $P$ are large, and their ratio $\alpha = P/N$ is finite; and the low-dimensional case where only $N$ is large, and $\alpha = 0$. The optimization problem we arrived at is generally nonconvex, so individual network solutions may depend on random weight initialization. Remarkably, however, the analytically derived steady-state firing-rate distribution is invariant to initialization. This distribution therefore, provides a reliable and direct link for comparing model predictions with experimental data.

The presence or absence of each stimulus was modeled by setting the corresponding components of $\boldsymbol{x}$ and $\boldsymbol{y}$ to 0 or 1. For example, the mismatch and match conditions for the $k$-th stimulus-pair correspond to,

$$\begin{aligned} (x^k, y^k) &= (1, 0) &\quad (x - \text{only mismatch condition}), \\ (x^k, y^k) &= (0, 1) &\quad (y - \text{only mismatch condition}), \\ (x^k, y^k) &= (1, 1) &\quad (\text{match condition}) \end{aligned}$$

Notably, our derivation of the network's response properties (SI §2.1) accounts also for potential complex (i.e., not one-to-one) pairings of the components $x_k$ and $y_k$, and for scenarios where the stimulus dimension differs across modalities. These results are further extended to apply in scenarios with associations between more than two stimuli (SI §2.3)

## Relationship between the network model in our work and recurrent autoencoders

We note that the network model we studied here and the objective function [Eqs. ((1), (2))] may appear similar to the definitions of regularized autoencoder models in the machine learning literature, particularly recurrent autoencoders[87]. We emphasize here that our recurrent network model differs from these models in several key aspects:

1. In our work, the neural dynamics themselves can be viewed as performing Bayesian inference of latent variables (SI §2.2). By contrast, in recurrent autoencoder models, the inferred latent variables are represented by a separate set of "auxiliary" variables which are not the neural activity itself[87]. Introducing a separate set of variables might limit the interpretability of this class of models in a neuroscientific context.

2. In recurrent autoencoders, the network is required to generate an accurate stimulus prediction at every time step[87]. By contrast, the network model we studied here imposes a less stringent constraint, requiring predictions to be generated only at steady-state.

3. The network model in this work can learn to generate predictions based on local plasticity rules (SI §2.3.2). The network structure emerging from these local plasticity rules is expected to yield neural representations and circuit structure distinct from those learned via the non-local back-propagation-through-time algorithm commonly used to train those recurrent autoencoder models.

In this work, we focus on the neural representations and circuit mechanisms that support multimodal, high-dimensional predictions in the brain, and link the network model to experimental data that probe the corresponding neural circuits.

## Geometry of representations of stimuli, predictions, and prediction-errors

Under the above assumptions, the steady-state neural response vector [Eq. (1)] can be expressed as,

$$r \propto \left[ a_x(\mu)x + a_y(\mu)y + \sqrt{\alpha} \cdot \text{noise} \right]_+. \tag{6}$$

This form is revealing, since the stimulus-specific, $\mu$-dependent vectors $a_x(\mu)$, $a_y(\mu)$ correspond to the directions along which the network encodes the stimuli in the $x$-only and $y$-only mismatch conditions. Eq. (6) also shows that, owing to the nonlinearity, the readout in the matched condition is not $a_x(\mu) + a_y(\mu)$. The geometry of representing stimuli in the match and mismatch conditions is illustrated in Fig. 1d. Changes to these vectors during training (i.e., $\mu$ increases) correspond to the learned structure of neural representations of stimuli and prediction-errors. We further note that the magnitude of the noise in Eq. (6) depends on the stimulus dimensionality $\alpha$, and thus it captures the interference between learned stimuli.

## Definition of balance level

At steady state, the balance level for neuron $i$ is defined as,

$$B_i = \left| \frac{I_i^F}{I_i^F - I_i^R} \right|. \tag{7}$$

Here, $I_i^F$ and $I_i^R$ are the feedforward and recurrent input currents to neuron $i$ at steady-state [Eq. (1)]. The balance level varies between

neurons and between stimuli, because the weights $w_i^k$ and $v_i^k$ are different for different neurons and stimuli (indexed by $i$ and $k$, respectively). The balance level distribution and its median shown in Fig. 2 were computed analytically (SI §3.3).

In scenarios where stimuli and neural responses vary over time (Figs. S6 and S7), the time-dependent balance level of neuron $i$ is defined similarly,

$$B_i(t) = \left| \frac{I_i^F(t)}{h_i(t)} \right|. \tag{8}$$

Here $I_i^F(t)$ is the time-dependent external input to neuron $i$. $h_i(t)$ is the voltage level of neuron $i$ at time $t$. At steady state, this definition coincides with Eq. (7).

## Extracting the optimal balance level from experimental data

**V-M experiment, ref. 20.** We calculated the trial-averaged voltage of all the recorded L2/3 neurons as a function of time (Fig. 3a). Voltage level of each neuron was measured with respect to its baseline. We sampled 50 voltage levels from all recorded neurons and all time points in the match and mismatch time windows (Fig. 3a), which were −0.1 to 0 s (match) and 0–0.1 s (mismatch). The time $t = 0$ corresponds to point at which the treadmill was decoupled from visual flow in virtual reality. We then computed the standard deviation over those 50 samples of the voltage level in the match and mismatch conditions. By taking the ratio of these standard deviations, we obtained a dimensionless quantity that has a direct analog in the model: the standard deviation of $h_i$ over neurons in the network in Eq. (1). Specifically, for $P = 1$, $\theta = 0$, we computed this ratio explicitly (SI §3),

$$\frac{\sigma_{\text{mismatch}}^2}{\sigma_{\text{match}}^2} = \frac{1}{2} \frac{\mu^2 + (1-\mu^2)(1+b/2)^2}{\mu^2 + \mu + (1-\mu^2)[1+b+(1-\mu)b^2/4]}. \tag{9}$$

We use $\mu = 0.97$ as the correlation value after training and fit this formula to the ratio obtained from data by adjusting the value of $b$. Using the best-fit value $b^\star$, we computed the median of balance level $B^\star$ in the network model (Fig. 3c).

**A-M experiment, ref. 12.** We calculated the trial-averaged firing-rates for all regular spiking neurons ($n = 815$) in the passive (mismatch) and movement (match) condition in two time windows: from $t = −0.1$ s to stimulus onset ($t = 0$), and from stimulus onset to $t = 0.06$ s (Fig. 3b). For every neuron, we calculated the change in its firing-rate between the two time windows in both conditions. We sampled 400 firing-rate change values from 815 neurons with replacement and calculated the average firing-rate change in the passive and movement conditions. We computed the equivalent quantity in the model, i.e., average of $\phi(h_i)$ over neurons in the network [Eq. (1)] in the match and mismatch conditions. For ReLU activation function, the ratio is also given by Eq. (9) and can be fit to the ratio obtained from the data by adjusting the parameter $b$. Again, we calculated the median of balance level $B^\star$ based on the best-fit value of $b^\star$. The fitting procedure for both experiments was repeated 100 times, giving the scatter plot of estimated $B^\star$ values (Fig. 3c).

## Definition of functional cell-types

We denote the steady-state voltage of neuron $i$ in the mismatch conditions as $h_i^x$ ($x$-only) and $h_i^y$ ($y$-only), and in the match condition as $h_i^{xy}$. To classify neurons into functional types, deviations of individual neurons' voltage response relative to the mean were compared to the standard deviation (denoted $\sigma$) of the steady-state voltage distribution. We evaluated $\sigma$ using the voltage distribution in the $x$-only mismatch condition after learning ($\mu = 0.97$).

A neuron $i$ is a representation (R) neuron for the $x$-stimulus if it is depolarized upon presentation of the stimulus $x$, i.e., its voltage

response in $x$-only mismatch condition is large, and its voltage responses in the match and mismatch conditions are similar. Mathematically,

$$h_i^x > \frac{\sigma}{2} \text{ and } |h_i^x - h_i^{xy}| < \frac{\sigma}{2}. \tag{10}$$

A similar criterion was used to identify $R$ neurons for the $y$-stimulus. A neuron $i$ is a prediction-error ($PE$) neuron if it signals the "mismatch" between $x$ and $y$, i.e., its voltage response in the $x$-only mismatch condition is large, and its voltage response in the match condition is small. Mathematically,

$$h_i^x > \frac{\sigma}{2} \text{ and } h_i^x - h_i^{xy} > \frac{\sigma}{2}. \tag{11}$$

Neurons meeting these criteria are referred to as positive PE neurons, because their activity increases when $x$ is presented but not expected (based on $y$). The activity of negative PE neurons increases when $x$ is not presented but is expected. In the model, E neurons have a centered (zero mean) distribution of voltages for $\alpha = 0$, therefore the threshold is applied to the voltage itself. When neural activity in the network is time-dependent (Figs. 5 and S6), the functional cell-types [Eqs. (10) and (11)] are computed at every time point based on the time-dependent voltage level $h_i(t)$ in different stimulus conditions, with a fixed threshold $\sigma$.

For excitatory neurons in the high-dimensional regime ($\alpha > 0$) and inhibitory neurons, since their voltage distribution has a non-zero mean, we used the centered voltage levels ($h_i^x$, $h_i^{xy}$) in the above criteria.

Note that neurons in the network may not belong to any of the those three classes (Fig. S3a). We computed the firing-rate statistics of neurons in the network analytically (SI §3 and 4), which allowed us to obtain the fraction of $R$ and $PE$ neurons for different values of $\mu$ and $\alpha$, shown in Fig. 4b, d. We further explored the effects of threshold level on the fraction of different functional types in Fig. S3b.

### Estimating functional segregation from responses to multiple stimuli from experimental data
We calculated the trial-averaged firing-rate change of each neuron in the match (active) and mismatch (passive) conditions, separately for each sound stimulus from our experimental data[13]. To calculate the segregation index for each type of probe sound, we restricted the analysis to neurons responsive in the passive condition to that probe sound and the learned (expected) sound. Responsive neurons were defined as those having firing-rate that was one half of the standard deviation above the mean firing-rate for the expected sound in the passive condition. Changing the threshold does not affect the results in Fig. 4e, f. For these neurons, we computed pairs of $\Delta$ values, defined as the difference between mismatch and match responses, for the probe and expected stimulus. The Pearson correlation coefficient between those $\Delta$ values was defined as the segregation index.

To estimate the similarity of the expected and probe stimuli, we computed individual neurons' trial-averaged firing-rate change following presentation of those stimuli in the passive condition from our experimental data[13] (the same time windows used in the A-M experiment, Fig. 3). For each animal, we considered population firing-rate vectors consisting of all its recorded neurons. Representation similarity was defined as the Pearson correlation of those vectors for pairs of auditory stimuli (expected and probe, Fig. 4f). We note that this similarity in the model is calculated from the activity of all neurons that are active in either the expected or probe stimuli in the passive condition.

### E/I network model
In the network with separate E and I neurons, the time-dependent voltages of E and I neurons are given by the following set of differential equations,

$$\frac{dh_i^E}{dt} = -h_i^E + \sum_{j=1}^{N_E} J_{ij}^{EE} \phi(h_j^E) + \sum_{j=1}^{N_I} J_{ij}^{EI} \phi_I(h_j^I) + I_i^E,$$
$$\tau_I \frac{dh_i^I}{dt} = -h_i^I + \sum_{j=1}^{N_E} J_{ij}^{IE} \phi(h_j^E) + \sum_{j=1}^{N_I} J_{ij}^{II} \phi_I(h_j^I) + I_i^I. \tag{12}$$

We assume that the activation function for inhibitory neurons is ReLU with zero threshold, $\phi_I(x) = \max\{x, 0\}$. Matching the E neurons' activity at steady state to the activity of neurons in our original network [Eq. (1)] gives constraints on the connectivity components and the feed-forward input (SI §5),

$$J^{EE} - J^{EI}(I + J^{II})^{-1}J^{IE} = J,$$
$$I^E - J^{EI}I^I = I^F. \tag{13}$$

Here $J$ and $I^F$ are the connectivity matrix and feedforward input used in Eq. (1). We further assume that the matrix $I + J^{II}$ is invertible. In general, there are many possible solutions $\{J^{EE}, J^{EI}, J^{IE}, J^{II}, I^E, I^I\}$ satisfying Eq. (13). We therefore identify a family of solutions. This continuum interpolates between the solution with private inhibition, where $J^{IE}$ is equal to the identity matrix; and solutions with an inhibitory internal prediction, where rows of $J^{IE}$ are given by the stimulus weight vectors (SI §5). Moreover, we show that up to a constant, the balance level defined earlier [Eq. (7)] is the same as the stimulus-specific, local component of the E/I balance level in the E/I network (SI §5).

We extended the definition of functional cell-types [Eqs. (10) and (11)] to I neurons. We note that here the average input to inhibitory neurons is not 0, so we subtracted the mean from the voltage level [$h$'s in Eqs. (10) and (11)] before applying the criteria on the deviations from the mean.

### Analyzing responses of regular spiking and fast spiking neurons
We estimated the connectivity structure parameter $\lambda_{EI}$ based on recordings of regular spiking and fast spiking neurons[12]. Using the same time windows as Figs. 3b and 4e, f, we calculated individual neurons' trial-averaged firing-rate change in the passive and movement conditions for the expected sound and the probe sound. Those firing-rate changes recorded in each animal form eight population vectors (regular/fast spiking, expected/probe sound, movement/passive). We calculated the Pearson correlation between population vectors under movement and passive conditions, giving four values for each animal, shown in Fig. 5d. The correlation values for presentation of the expected sound were regarded as "after learning", while correlation values for presentation of the probe sound that was not associated with the lever press were regarded as "before learning".

### Hierarchical recurrent network model
In the hierarchical network model, each neuron belongs to one of three modules, indicated by superscripts in the equations governing neural activity,

$$\frac{dh_i^1}{dt} = -h_i^1(t) + b_1\left(\sum_j J_{ij}^1 \phi(h_j^1(t)) + \sum_k W_{ik}^1 x_k + \sum_{k'} V_{ik'}^1 \phi(h_{k'}^2(t))\right) \tag{M1}$$

$$\frac{dh_i^2}{dt} = -h_i^2(t) + b_2\left(\sum_j J_{ij}^2 \phi(h_j^2(t)) + \sum_k W_{ik}^2 \phi(h_k^1(t)) + \sum_{k'} V_{ik'}^2 \phi(h_{k'}^3(t))\right) \tag{M2}$$

$$\frac{dh_i^3}{dt} = -h_i^3(t) + b_3\left(\sum_j J_{ij}^3 \phi(h_j^3(t)) + \sum_k W_{ik}^3 \phi(h_k^2(t)) + \sum_{k'} V_{ik'}^3 y_{k'}\right) \tag{M3}$$

$$\tag{14}$$

The definitions of feedforward and recurrent connectivity are generalizations of the single-module network. Specifically, $J_{ij}^l = -\frac{b_i}{N}\sum_k(W_{ik}^l W_{jk}^l + V_{ik}^l V_{jk}^l)$. These neural dynamics minimize the bidirectional prediction errors that are local to each module in the objective function (Fig. S11a, right; SI §2.3). Moreover, the model can be extended to a hierarchical network with an arbitrary number of layers (SI §2.3).

In Fig. S11, the above model is compared with the unidirectional predictive processing model with three modules. The neural activity in each module in this unidirectional model are governed by,

$$\frac{dh_i^1}{dt} = -h_i^1(t) + b_1\left(\sum_j J_{ij}^{1u}\phi(h_j^1(t)) + \sum_k W_{ik}^1 x_k + \sum_{k'} W_{ik'}^2\phi(h_{k'}^2(t))\right) \quad \text{(M1)}$$

$$\frac{dh_i^2}{dt} = -h_i^2(t) + b_2\left(\sum_j J_{ij}^2\phi(h_j^2(t)) + \sum_k W_{ik}^2\phi(h_k^1(t)) + \sum_{k'} V_{ik'}^2\phi(h_{k'}^3(t))\right) \quad \text{(M2)}$$

$$\frac{dh_i^3}{dt} = -h_i^3(t) + b_3\left(\sum_j J_{ij}^{3u}\phi(h_j^3(t)) + \sum_k V_{ik}^2\phi(h_k^2(t)) + \sum_{k'} V_{ik'}^3 y_{k'}\right) \quad \text{(M3)}$$

$$\quad (15)$$

where $J_{ij}^{1u} = -\frac{b_i}{N}\sum_k W_{ik}^1 W_{jk}^1 - \delta_{ij}$ and $J_{ij}^{3u} = -\frac{b_i}{N}\sum_k V_{ik}^3 V_{jk}^3 - \delta_{ij}$. These neural dynamics minimize the objective function shown in Fig. S11a, left, where M1 and M3 generate predictions of their corresponding stimulus inputs and M2 generates predictions of activity in M1 and M3.

## Statistical tests

In Figs. 3c, 4f, and 5d, we used two-sided, unpaired $t$-tests. $* = p < 0.05$ and $*** = p < 0.0005$.

## Reporting summary

Further information on research design is available in the Nature Portfolio Reporting Summary linked to this article.

## Data availability

No new experimental data was collected in this study. Source Data files are provided for all figures.

## Code availability

Computer code to reproduce model simulations is available in the Github repository: https://github.com/BinW3233/HDPC_code.git.

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

## Acknowledgements

The authors thank S. Azizpour Lindi, D. Bambah-Mukku, A. Finkelstein, E. Mukamel, I. Nelken, and C.-Y. Su for useful conversations, and I. Nelken for comments on a previous version of the manuscript. This work was supported by DARPA grant D21AP10162-00 (J.A.), DOE grant DE-SC0022042 (J.A.), NIH grants 1R01-NS135853 (J.A.), K99-DC020770 (N.J.A.), 1R01-DC018802 (D.M.S). B.W. thanks NSF grant DBI-2229929, Simons Foundation, Swartz Foundation, Kavli Foundation, and UCSD Friends of the International Center for support. D.M.S. is a New York Stem Cell Foundation–Robertson Neuroscience Investigator. J.A. and B.W. thank the Kavli Institute for Theoretical Physics (KITP), Tel Aviv University, and ICERM at Brown University for hospitality during summer and fall 2023. KITP is supported by NSF grant PHY-1748958 and the Gordon and Betty Moore Foundation Grant No. 2919.02. ICERM is supported by NSF grant DMS-1929284.

## Author contributions

Developed the project and modeling approach, B.W., J.A. Solved, analyzed, and simulated model, B.W. with inputs from J.A. Designed and performed experiments, N.J.A., D.M.S. Designed and performed data analysis, B.W., J.A. with inputs from N.J.A., D.M.S. Wrote paper, B.W., J.A. with inputs from N.J.A., D.M.S. Supervised the project, J.A.

## Competing interests

The authors declare no competing interests.
