## [Transparent Peer Review file · Nature Communications]

Desegregation of neuronal predictive processing

Corresponding Author: Professor Johnatan Aljadeff

Version 0:

Reviewer comments:

Reviewer #1

(Remarks to the Author)

The authors have satisfactorily addressed my main concerns. They have tempered their claims about offering a general theory and now clearly state the restricted stimulus regime used, while also clarifying robustness to some non-orthogonal inputs. Terminology is now aligned with or explicitly justified against field standards. They have also strengthened the empirical link by framing their key, falsifiable prediction, that neurons are not segregated into representation and prediction-error populations, and showing consistency with available data, while acknowledging the limits of their analyses. Although the manuscript still lacks the kind of rigorous, quantitative model–data validation I originally hoped for, the authors have appropriately reframed these as directions for future work. With these changes, the manuscript no longer overstates its scope or generality, and I am satisfied that the current version makes a clear and useful contribution. I support publication, noting that more quantitative tests will be important in subsequent work.

(Remarks on code availability)

The code provided is appropriate.

Reviewer #2

(Remarks to the Author)

The authors have responded sufficiently to my comments, and I think the paper is appropriate for publication at Nature Communications.

(Remarks on code availability)

Reviewer #3

(Remarks to the Author)

In my original review, I stated that the key problem with the paper by Wang et al is that it fails to properly build on the state of the art and does not clearly distinguish between what is new and what is known. Although I find that the revised paper has improved in certain aspects, many of the original problems persist. This is not simply a matter of giving proper credit. It simply should not be left to the reader to have to work out what is new in a paper.

In addition, I find there is a general problem with the clarity of the writing. Similar to ref #1, I note that one big problem is that the authors are not clear in their use of terms. For instance, they compare concepts (e.g. prediction error neuron) using different definitions, which tends to obfuscate rather than clarify things. Here are the main points:

(1) The authors keep insisting that they are investigating a new network model. Sorry, but that is just not correct. The authors use classical sparse coding and predictive coding models and apply them to multi-modal sensory inputs. These models are classical for a reason: there is a vast literature about them.

In their reply, the authors emphasize that they rectify the firing rates, which the original Olshausen & Field model (from 1996) did not. Correct. However, this change is hardly novel and has been repeatedly used and studied before. For instance, here are two papers that use a model that uses rectification and is mathematically identical to the author's model: Rozell, ...,

Olshausen (2008) Neural computation; Zhu & Rozell (2013) PLOS CB.

Similarly for the predictive coding model. Indeed, Rao & Ballard (from 1998) do not rectify the firing rates, but later research has done so.

The authors furthermore state: "It is well-known that implementing the learning rules proposed in these classical network models only guarantees a local (not global) maximum of the objective function, and there is no guarantee that the same solution will be found in each repetition of the training procedure. In contrast, we show..."

Indeed, the loss functions are not convex. But since the authors use the same model, with the same loss function, and the same learning rules, they suffer from the same problem. They emphasize that they show convergence of *statistics* of firing rates, but that's a different issue, and, frankly, was already a point made in the original papers (e.g. sparse firing rate distributions, statistics of receptive fields).

(2) The authors point out that the desegregation of prediction error neurons is a novelty of their model. It seems to me now that here is what happens: the original predictive coding studies gave the name 'prediction error neuron' to specific computational units that compared top-down predictions to bottom-up sensory inputs. In contrast, Wang et al call 'prediction error neuron' any neuron that shows a mismatch response. These definitions are obviously not the same!

The confusion then arises because the authors claim that there is a difference in their model compared to the previous models. However, the difference seems to be simply in the definition of what constitutes a prediction error neuron.

This should be made much clearer, because otherwise the authors are comparing apples and oranges.

The authors also claim that their predictive coding model is new and different because it has 'bidirectional' predictions. That seems like obfuscation. What the authors really seem to do is the following: use a predictive-coding model a la Rao & Ballard (but with rectification, so maybe more similar to Spratling or Friston) with two stages. The first stage receives inputs from the two sensory modalities (they call it M1 and M3), the second stage (called M2) computes the predictions. That's it.

The confusion then arises because the authors reorder the stages into a different type of hierarchy, M1-M2-M3. Using this relabeling of the stages, the authors then observe that the 'middle module' generates prediction errors for both the 'lower' (M1) and 'higher' module (M3). They label this type of prediction 'bidirectional' and state that it is different from the original predictive coding model.

(3) Move from balance to unbalanced states. I am generally ok with their answer, but I note that they are citing the wrong papers from the Deneve lab. The right paper to cite is: Brendel et al, 2020, PLOS CB.

(4) There are multiple other instances where the text ranges from sloppy to raising false expectations to incorrect. For instance,

line 40: "... multiple brain circuits outside of the mammalian cortex exhibit predictive coding... It is unknown whether these neural circuits employ similar or entirely different strategies for predictive processing compared to the mammalian cortex" - In the introduction, all kinds of interesting questions are raised, but they are then not addressed in the paper.

line 57: "these studies typically focus on predicting a small number of stimuli within a single sensory modality" - studies focusing on visual natural stimuli are hardly using a 'small number of stimuli'

line 64: "Another major current gap from both experimental and modeling perspectives is predictive processing in high-dimensions" - the original predictive coding paper use visual image patches (200-300 input dimensions). The stimuli used by the authors are much simpler (and lower-dimensional) by comparison.

line 72: "We address these questions by developing a mathematical framework" - not clear what the mathematical framework is. It's certainly not the network model.

...

(Remarks on code availability)

Version 1:

Reviewer comments:

Reviewer #3

(Remarks to the Author)

All of my concerns have been adequately addressed by the authors.

(Remarks on code availability)

Reviewer #1 (Remarks to the Author):

The authors have satisfactorily addressed my main concerns. They have tempered their claims about offering a general theory and now clearly state the restricted stimulus regime used, while also clarifying robustness to some non-orthogonal inputs. Terminology is now aligned with or explicitly justified against field standards. They have also strengthened the empirical link by framing their key, falsifiable prediction, that neurons are not segregated into representation and prediction-error populations, and showing consistency with available data, while acknowledging the limits of their analyses. Although the manuscript still lacks the kind of rigorous, quantitative model–data validation I originally hoped for, the authors have appropriately reframed these as directions for future work. With these changes, the manuscript no longer overstates its scope or generality, and I am satisfied that the current version makes a clear and useful contribution. I support publication, noting that more quantitative tests will be important in subsequent work.

Reviewer #1 (Remarks on code availability):

The code provided is appropriate.

We sincerely thank the Reviewer’s constructive comments, helping us strengthen the conclusion of the paper, and for finding our contributions ‘clear and useful’. We note that in the current version of the manuscript, we have carried out statistically rigorous experimental test on the model predictions in Fig. 3. As the Reviewer’s suggested, future work will involve additional quantitative model-data comparisons.

Reviewer #2 (Remarks to the Author):

The authors have responded sufficiently to my comments, and I think the paper is appropriate for publication at Nature Communications.

We thank the Reviewer’s comments and support for our work.

Reviewer #3 (Remarks to the Author):

In my original review, I stated that the key problem with the paper by Wang et al is that it fails to properly build on the state of the art and does not clearly distinguish between what is new and what is known. Although I find that the revised paper has improved in certain aspects, many of the original problems persist. This is not simply a matter of giving proper credit. It simply should not be left to the reader to have to work out what is new in a paper. In addition, I find there is a general problem with the clarity of the writing. Similar to ref 1, I note that one big problem is that the authors are not clear in their use of terms. For instance, they compare concepts (e.g. prediction error neuron) using different definitions, which tends to obfuscate rather than clarify things. Here are the main points:

We sincerely thank the Reviewer once again for their time and effort in re-assessing our work and for providing constructive feedback to improve the clarity of the manuscript. In the revised manuscript, we added simulation results to further clarify the novelty in our work by comparing it directly to previous modeling studies. Those new results appear in Fig. 4d, right; Fig. 7g right, Fig. S11 and lines 297-300, 520-536, 547-549. We also elaborated on the main contributions of our work in the abstract, introduction, and discussion (lines 13-14, 69-74, 82-83, 94-98, 567-568). Additionally, we expanded the discussion and implications of the precise definitions we used and their connection to existing literature (lines 262-271, 621-624, 631-636). Below, we provide a detailed explanation of these revisions.

(1) The authors keep insisting that they are investigating a new network model. Sorry, but that is just not correct. The authors use classical sparse coding and predictive coding models and apply them to multi-modal sensory inputs. These models are classical for a reason: there is a vast literature about them.

In their reply, the authors emphasize that they rectify the firing rates, which the original Olshausen & Field model (from 1996) did not. Correct. However, this change is hardly novel and has been repeatedly used and studied before. For instance, here are two papers that use a model that uses rectification and is mathematically identical to the author’s model: Rozell, ..., Olshausen (2008) Neural computation; Zhu & Rozell (2013) PLOS CB.

Similarly for the predictive coding model. Indeed, Rao & Ballard (from 1998) do not rectify the firing rates, but later research has done so.

We agree with the Reviewer that existing studies also employ rectified firing rates. Yet, there are fundamental differences between our study and previous literature that we clarify further here.

In the manuscript, the term ‘model’ refers holistically to a system with three components investigated jointly: the stimulus inputs, the network dynamics, and the learning rule that determines the connectivity. While it is true that the network dynamics and learning rules we use are similar to classical sparse coding and predictive coding literature on neural processing of natural images, the stimulus inputs in our work are fundamentally different. We note that, since the network connectivity is jointly determined by the inputs and the learning rule, the network models we investigated for multi-modal and high-dimensional inputs have different connectivity from previous works. Our choice of the input structure is motivated by recent experiments on multi-modal associative learning (e.g., combinations of sensory and motor inputs or sensory inputs from different modalities) and allows us to investigate questions beyond inputs that probe a single sensory modality. For example, the nearly orthogonal stimulus pairs provide a direct experimental analogue for conditions in which animals learn sensory-motor associations between multiple familiar but unrelated stimuli. Moreover, the dimensionality of these multi-modal inputs can be systematically varied in the model, enabling us to study neural responses and circuit computations across environments with different degrees of complexity. This more general stimulus structure used in the model allows us to directly test the predictive processing hypothesis in multi-modal settings using recent experimental data.

While the properties of the stimulus and how they affect the circuit structure are the new components of the model we studied, we emphasize that the contributions of our work go beyond including a generalized paired input (x, y) in the model equations. We believe that our main contributions lie in the *new results* that characterize the learned neural representations and circuit structure for multi-modal, high-dimensional inputs, and that demonstrate how these findings are linked to experimental data.

To be more specific about the new results in our work, for the multi-modal stimuli, we derived analytical formulas that characterize the responses of *all* neurons in the nonlinear network [Eqs. (S67-S69), a simplified form is shown in Eq. (6) in the main text]. To our knowledge, these are the first analytical results that describe how each neuron’s response and the inputs it receives depend jointly on the multi-modal stimulus condition (x^k, y^k) , the dimensionality of the stimulus set (α) , the sparsity and regularization parameters (b, θ) and the learning stage (μ) , or equivalently the number of training epochs). This analytical advance enables us to efficiently and systematically investigate how stimulus dimensionality and different types of mismatch signals (between multiple stimulus pairs) shape neural

response statistics and degrees of balance in a multi-modal setting (Fig. 2 and Fig. 4), and to relate these effects directly to recent experimental data (Fig. 3 and Fig. 4e,f).

An interesting phenomenon revealed by our analysis and not previously appreciated in the literature is that the prediction-error and stimulus representations are mixed at the single-neuron level. These were established based on analytical results pertaining to the steady-state neural responses in a single-module network, and then expanded upon along multiple directions: transient network dynamics evoked by paired multi-modal inputs (Fig. 5); differences between responses of inhibitory and excitatory neurons (Fig. 6); and module-specific responses in a hierarchical network (Fig. 7). These latter results are especially timely given the emergence of experimental technologies for population-level, temporally resolved, cell-type-specific and region-specific neural recordings.

These new analytical results, the new phenomena we identified in neural responses for multi-modal high-dimensional stimuli, and the comparisons to recent experimental data on multi-modal associative learning together constitute the main innovations of our work. None of these findings have been revealed in the aforementioned work, which focused on the visual system. In fact, some of the questions we address here are not even well-defined in the context of a network processing inputs from a single sensory modality. In our study, the dimensionality of a multi-modal stimulus-set can be systematically varied in experiments to test the model predictions we propose. It is unclear how to perform an analogous manipulation in the context of a network processing natural images, since the presentation of specific images in an experiment is not typically dependent on specific motor actions or other sensory modalities.

We agree that describing the model in our work as ‘a new network model’ was indeed potentially confusing. In the revised manuscript, we have removed this wording, emphasized that the network model we study follows from the seminal sparse-coding and predictive-coding literature, and clarified that the novelty of our work lies in the new insights it provides into multi-modal, high-dimensional responses in sensory brain regions (line 55 82-83, 97-98, 133-134, 567-568). These insights arise from extending the results obtained in previous studies to a new setting with paired multi-modal stimuli.

We respectfully disagree with the Reviewer’s characterization that in our work we simply ‘*use classical sparse coding and predictive coding models and apply them to multi-modal sensory inputs.*’ This seems to suggest that our results follow trivially from prior studies. First, our work also incorporates locomotion signals arising from motor actions, not merely multi-modal sensory inputs (Fig. 3b, Fig. 4e). Second, we believe that understanding neural responses to sensory modalities other than vision is an equally important neuroscientific question as understanding responses to visual stimuli, which was the focus of the earlier work cited by the Reviewer. Moreover, our study specifically investigates *multi-modal responses* within a single primary sensory area (V1 or A1), as observed in recent experiments. Through a combination of analytical calculations, numerical simulations and analysis of experimental data, we examined how such multi-modal responses are organized within nonlinear neural circuits (e.g., nonlinear mixed selectivity to multi-modal sensory and prediction-error signals) and how stimulus dimensionality, transient stimulus dynamics and other related stimulus properties shape these responses. Together with the recent experimental findings, our results represent an important conceptual advance beyond the long-standing assumption that primary sensory areas are specialized exclusively for the statistics of sensory signals of a single sensory modality. As we stated in the Introduction, the central questions motivating our work are how multi-modal neural responses are organized in a network that performs predictive processing, and how these responses change as the stimulus complexity (dimensionality) is systematically increased. In the revised manuscript, we have clarified these points and highlighted the conceptual contributions of our study (line 69-74, 567-568).

We provide a detailed comparison between our work and the two additional studies mentioned by the Reviewer:

- **Rozell, Johnson, Baraniuk and Olshausen (2008) *Neural Computation*.** This study analyzed neural dynamics (the local competition algorithm) that minimizes a prediction error objective with sparsity regularization. Different forms of regularization were shown to correspond to different nonlinear single-neuron activation functions, including a rectified linear transformation. This part of the result is similar to the neural dynamics studied in our work. However, our analysis goes further by providing explicit analytical expressions characterizing how individual neurons respond to stimuli after training. Moreover, the same analytical results also describe how neural responses evolve during training under the learning rule, which was not addressed in Rozell et al. These advances allow us to connect our model directly to experimental data showing how training shapes neural responses.
- **Zhu & Rozell (2013) *PLoS Computational Biology*.** This work studied a classical sparse coding model and used it to explain non-classical receptive fields in V1 in response to the image patches. Their regularization term in the objective corresponds to a nonlinear activation function ($T_\lambda(\cdot)$ in Zhu & Rozell) that can take both positive and negative values. The positivity of the neural activity is then enforced manually. In contrast, our model uses a nonlinear activation function that ensures positivity directly, leading to a mixed L_1 and L_2

regularization [Eq. (S5)], rather than the purely L_1 regularization used in Zhu & Rozell. Moreover, Zhu & Rozell focuses on a subset of neurons in the network that exhibit non-classical receptive fields, while our work considers and provides analytical results pertaining to the responses of *all* neurons in the network.

Thus, we believe that the models used in these two papers are not mathematically identical to ours. Neither study examines multi-modal stimulus inputs as in our setting, nor do they investigate the effects of systematically increasing stimulus dimensionality, transient network dynamics, or inhibitory neuron responses, all of which are key components for linking multi-modal, high-dimensional neural representations in the model to recent experimental data. In the revised manuscript, we have added a concise version of these comparisons for readability and have highlighted the significance of our contributions to improve clarity (line 54, 133-134, 692-695).

The authors furthermore state: "It is well-known that implementing the learning rules proposed in these classical network models only guarantees a local (not global) maximum of the objective function, and there is no guarantee that the same solution will be found in each repetition of the training procedure. In contrast, we show..."

Indeed, the loss functions are not convex. But since the authors use the same model, with the same loss function, and the same learning rules, they suffer from the same problem. They emphasize that they show convergence of *statistics* of firing rates, but that's a different issue, and, frankly, was already a point made in the original papers (e.g. sparse firing rate distributions, statistics of receptive fields).

We apologize for not sufficiently clarifying this issue in our previous reply and in the earlier version of the manuscript. We agree with the Reviewer that the loss function of the model in our work [Eq. (S6)] is jointly non-convex as a function of the firing rates and weights, similarly to classical predictive coding and sparse coding models. Since a non-convex objective may have multiple local minima, gradient-based optimization can in principle converge to different network solutions depending on random weight initializations. These different solutions may correspond to substantially different neural response properties and circuit structure. This poses a potential challenge for using the model to generate predictions for experimental data, since different network solutions could yield different predictions. To address this issue, our goal in this work is to identify consistent properties across all these network solutions, when trained on multi-modal stimuli with random weight initializations.

By analyzing the neural responses in the network during training mathematically, we showed that in the limit of large network-size ($N \gg 1$), both the firing rate distribution and the distribution of inputs to each neuron in the network become independent of random weight initialization. Moreover, these distributions are given explicitly by the analytical formulas we derived in Eqs. (S67-S69): Eq. (S67) gives the firing rate/voltage distributions, which depend on unknown parameters $q, \hat{q}, \hat{q}', t^k, s^k$ that need to be solved from Eq. (S68-S69). In other words, these distributions represent the consistent properties shared by all the network solutions obtained via the gradient-based learning rule on the non-convex loss function. These analytical results were further confirmed by numerical simulations where the firing rate statistics (Fig. 1c,e) and the fraction of different functional neuronal-types (Fig. 4b,d) were compared to the theory. Guided by these insights from the analytical theory, we therefore compare these distributional properties between models and experimental data (Fig. 3, Fig. 4e,f, Fig. 6d), rather than individual neuron responses.

We agree with the Reviewer that the original papers of sparse/predictive coding also examined firing-rate distributions and receptive-field statistics empirically. However, convincingly demonstrating that these distributional properties are preserved across all the network solutions would require training a vast number of networks with different random weight initializations to adequately sample the parameter space (whose dimension is the total dimension of all weight vectors): due to the curse of dimensionality, the number of required network samples grows exponentially with network size, making this infeasible for large networks. The situation would be even worse given our goal to systematically vary the dimensionality of the learned stimulus ensemble and compare the resulting differences in the network solutions, which would require re-training a different network for stimulus ensembles with different dimension.

Here using analytical calculations, we were able to show explicitly that these distributional properties converge and are consistent, in the large network limit, across all network solutions and for any stimulus dimensionality. Moreover, instead of training an astronomical number of networks to convincingly sample from these distributions, these distributions are summarized in a compact way by the analytical expressions [Eq. (S67-S69)]. We believe that this analytical theory offers an effective and powerful framework for characterizing neural representations and the structure of circuits implementing predictive processing of multi-modal, high-dimensional stimuli. In the revised manuscript, we have elaborated on this point to help readers appreciate the value of our analytical results (line 136-139, 722-726). We thank the Reviewer for raising this issue and for helping us sharpen the main messages of the paper.

(2) The authors point out that the desegregation of prediction error neurons is a novelty of their model. It seems to me now that here is what happens: the original predictive coding studies gave the name 'prediction error neuron' to

specific computational units that compared top-down predictions to bottom-up sensory inputs. In contrast, Wang et al call 'prediction error neuron' any neuron that shows a mismatch response. These definitions are obviously not the same!

The confusion then arises because the authors claim that there is a difference in their model compared to the previous models. However, the difference seems to be simply in the definition of what constitutes a prediction error neuron.

This should be made much clearer, because otherwise the authors are comparing apples and oranges.

We thank the Reviewer for raising this important issue and for identifying the source of confusion in the earlier version of the manuscript. There is indeed a difference between our definition of prediction-error neurons and the definition used in the classical predictive coding literature (e.g., Rao & Ballard). It is important to note that similar discrepancies exist between the modeling literature and the experimental literature regarding how prediction-error neurons are defined. Below, we first elaborate on this issue and explain the motivation behind the definition used in our work, and then clarify the novelty of our results.

Two types of network models exist in classical sparse and predictive coding literature (Fig. R1): one type of model originates from sparse coding papers (Olshausen & Field 1996, including the two follow-up papers mentioned by the Reviewer; **Model A in Fig. R1**). This is a recurrent neural network model where the recurrent connections between neurons are given by inner products of the readout weights, similar to the model in our work. Another type of model originated from the seminal contribution of Rao & Ballard (1999) and the papers that followed (e.g., Spratling and Friston; **Models B1/B2 in Fig. R1**). Here each layer of the network consists of two types of neuronal populations, one population maintains the current estimate of the input signal (also referred to as 'value units' in later literature), and another population detects errors and computes the difference between current estimate (bottom-up inputs) and top-down signals, referred as to prediction-error neurons (or 'error units') in Rao & Ballard (1999). The error units have one-to-one connection to those value units. As the Reviewer suggested, according to this definition of prediction error neurons, the sparse coding model (Model A) does not have any prediction error neurons by definition.

The most direct approach experimentally identify the error units proposed by the Rao & Ballard's model, would be to look at the connectivity between neurons in the corresponding sensory area (in this context, the layer 2/3 of primary sensory area). Currently, such a dense connectomic reconstruction is not available.

A more tractable approach involves analysis of experimental measurements of neural responses. Therefore, recent experimental work along this direction used a criterion [from Keller & Mrsic-Flogel (2018) *Neuron*, and subsequent papers such as refs. 12,13,16,18,20 in our manuscript's bibliography], which leverages neural responses in match and mismatch conditions. This is the prediction error neuron definition used in our work, as our work is motivated by these recent experimental data on multi-modal inputs. We believe this definition is not an obfuscation, but rather a useful tool for connecting mathematical models to experimental data. Moreover, the error units in the original predictive coding studies would also meet the criteria and be identified as prediction-error neurons by this definition.

With the two definitions of *prediction-error neurons* clarified, next we clarify why we believe our findings of functional desegregation of prediction-error representations are indeed novel results:

We note that Olshausen & Field and Rao & Ballard noticed that, *for unimodal inputs* (i.e., eliminating the y input in Fig. R1), the two types of models are in fact mathematically equivalent to each other. However, the two types of models differ when considering multi-modal stimuli (Fig. R1). Our work focused on Model A, which is the sparse-coding-based model driven by multi-modal inputs. On the other hand, the model with dedicated error-units has two variants, depending on whether the y inputs are top-down (Model B1) or bottom-up (Model B2). Our focus in the manuscript is on the convergence of bottom-up auditory signals and top-down motor signals on circuits in the auditory cortex, so we consider the comparison between Model A and Model B1.

One key difference between the two types of models (Model A vs. B1/B2), is the implementation of prediction error computation: in Model A, the prediction error is computed through recurrent connections, and is distributed across the network. No single neuron or any subset of neurons is dedicated to computing the error. Indeed, as we showed in our work, individual neurons have mixed representations of prediction-errors and the stimuli after training. On the other hand, in Model B1 and Model B2, a dedicated neuronal population (the error units) computes prediction-errors. Based on the criteria for identifying prediction-error neurons in the experimental literature, this dedicated population will always be classified as prediction-error neurons and will not have mixed representation. Such differences in neuronal implementation manifest in the response properties of experimentally recorded neurons:

Using the criteria from the experimental literature, Models B1 and B2 would predict the existence of a fixed fraction of neurons that are always classified as signaling prediction errors (i.e., *dedicated* prediction-error neurons). The functional properties of this subset of neurons is independent of the stimulus dimensionality, since by design their

Figure R1: Comparisons between different sparse and predictive coding models for multimodal inputs. Neural implementations of predictive error computation are different in Model A and in Model B1/B2.

activity always signals prediction-errors. By contrast, we examined the neural responses in Model A in the multimodal scenario and showed that neural responses are functionally desegregated, which we quantified as follows:

- When the stimulus dimension $P \geq 2$, a neuron that encodes a prediction-error for a specific stimulus dimension can encode another stimulus dimension irrespective of whether it is predicted or not (Fig. 4d, left).
- For high-dimensional stimuli (large P), most of the neurons in Model A have mixed error and stimulus representations (Fig. S4B).
- To further strengthen this result, in the revised manuscript we added an analysis demonstrating that neurons that encode prediction-errors exclusively (i.e., do not have responses that are classified as ‘stimulus representation’ for any of the P stimulus dimensions) are *exponentially rare* with increasing stimulus dimension P (Fig. 4d, right).

This key feature of our model—a functionally desegregated representation of stimuli and prediction-errors—was further supported by experimental data (Fig. 4e,f). This difference between Model A and Models B1/B2 was the difference we referred to in the earlier version of our manuscript. To our knowledge, such a mixed representation of stimulus and error representation in Model A and its dependency on stimulus dimensionality has never been investigated before, especially for multi-modal inputs.

In the revised manuscript, we now expanded the explanation of the differences in definitions of prediction-error neurons, differences in neural implementation of prediction-error computations in different models (line 262-271, 634-636). For readability, we did not include Fig. R1 in the manuscript, but we would be happy to do so if the Reviewer thinks it would improve the paper. We also highlight that the novelty of our findings stems from characterizing multi-modal, functionally mixed responses specifically in network models like Model A with multimodal inputs (line 631-633). Beyond the new panel we added to show that the fraction of dedicated prediction-error neurons decreases as stimulus dimension in the single module network (Fig. 4d, right, line 297-300), we carried out a similar analysis in the hierarchical model (Fig. 7g, right, line 547-549), where we show that pure prediction-error neurons become exponentially rare for increasing P in the middle module (M2, integrating bottom-up and top-down inputs). Based on this, we believe that our work represents an important advance towards validating mathematical network models of predictive processing based on cellular-resolution neural recordings.

We acknowledge that the experimental data we show do not rule out the possibility that dedicated prediction-error neurons may exist in other cell types not recorded in our dataset. Future recordings that target a broader diversity of cell types will be required to test this possibility. We have added this clarification in the Discussion of our manuscript (line 634-636).

The authors also claim that their predictive coding model is new and different because it has ‘bidirectional’ predictions. That seems like obfuscation. What the authors really seem to do is the following: use a predictive-coding model a la Rao & Ballard (but with rectification, so maybe more similar to Spratling or Friston) with two stages. The first stage receives inputs from the two sensory modalities (they call it M1 and M3), the second stage (called M2) computes the predictions. That’s it.

The confusion then arises because the authors reorder the stages into a different type of hierarchy, M1-M2-M3. Using this relabeling of the stages, the authors then observe that the ‘middle module’ generates prediction errors for both the ‘lower’ (M1) and ‘higher’ module (M3). They label this type of prediction ‘bidirectional’ and state that it is different from the original predictive coding model.

We thank the Reviewer for suggesting another way to think about the three stages in the hierarchical model. However, we believe that the bidirectional hierarchical network model in our work is indeed different from models in previous literature (including Rao & Ballard). In the single-module network we studied, the differences between our work and previous work stemmed from the incorporation of a multi-modal, high-dimensional stimulus-set, as discussed in response to previous comments. When extending the results to a hierarchical network with multiple modules, additional differences arise from considering a different form of the loss function that the network minimizes. We further note that each module in the hierarchical network model is a sub-network similar to Model A in Fig. R1. To clarify these differences, we added a new supplementary figure in the revised manuscript (Fig. S11), comparing our model to previous models. For convenience, this new figure is included in this document, Fig. R2:

In the left column of Fig. R2a, we formulated the multi-stage network model suggested by the Reviewer. In this model, M1 and M3 receive sensory inputs and the network minimizes the corresponding prediction errors, while M2 generates predictions of the activity of neurons in M1 and M3. These terms are shown in the objective function on the left side of the first row in Fig. R2a. Since the prediction signals in this model flow in only one direction (M2→M1, M3→sensory inputs), we refer to it as the ‘unidirectional predictive processing model’. The objective function of the hierarchical model studied in our work is shown on the right side of Fig. R2a. In addition to all the prediction-error terms present in the unidirectional model, this objective includes two additional terms: M1 and M3 also generate ‘backward’ predictions of the activity of neurons in M2. Since the predictions in this model flow in both directions, we refer to it as the ‘bidirectional predictive processing model’.

Due to the differences in their objective functions, the two models have different neural dynamics and network connectivity. In the unidirectional model, following Rao & Ballard, we assume that the voltage level of each neuron is driven by the negative gradient of the energy function ($\frac{dh_i}{dt} \propto -\frac{\partial E}{\partial r_i}$), resulting in a network shown on the left side of Fig. R2a. By contrast, in the case of the bidirectional model, we assumed that neural dynamics are driven by the negative gradient of the local errors [Eqs. (S28-S29)], leading to a different network structure shown on the right. The major differences between the two networks lie in the recurrent connectivity within M1 and M3 and the feedforward projections from M1/M3 to M2 (highlighted in Fig. R2a).

These differences lead to distinct neural response patterns to multi-modal inputs. In the unidirectional model, M3 exhibits minimal cross-modal responses throughout training (Fig. R2b) in the x -only mismatch condition. This contrasts with the bidirectional model (Fig. 7c), where neurons in M3 develop robust selectivity to the cross-modal stimulus x . Similar effects are also observed when we perform binary decoding for stimulus x in M3 in the two models. After training, the decoding performance for stimulus x based on responses of M3 neurons in the bidirectional model is higher than in the unidirectional model across different stimulus conditions and noise levels (Fig. R2c). The robustness of cross-modal responses in M3 during training and the superior decoding performance in M3 are consistent with recent experimental data showing cross-modal responses in visual and auditory regions. These results suggest that the bidirectional prediction model may provide a more accurate account of the multi-modal predictive computation in biological neural circuits.

In addition to the newly added figure (Fig. S11), we have revised the Methods and main text (line 502-503, 521-537, 858-861) to further clarify these differences. We thank the Reviewer for bringing up this issue, which helped us strengthen the correspondence of our model and empirical observations, as well as the presentation of our results.

(3) Move from balance to unbalanced states. I am generally ok with their answer, but I note that they are citing the wrong papers from the Deneve lab. The right paper to cite is: Brendel et al, 2020, PLOS CB.

We thank the Reviewer for pointing out this missing literature. We now added to the revised manuscript (line 58, 194). We believe that other papers we cited from Deneve’s lab are also relevant to our study.

(4) There are multiple other instances where the text ranges from sloppy to raising false expectations to incorrect. For instance,

line 40: “... multiple brain circuits outside of the mammalian cortex exhibit predictive coding... It is unknown whether these neural circuits employ similar or entirely different strategies for predictive processing compared to the mammalian cortex” - In the introduction, all kinds of interesting questions are raised, but they are then not addressed in the paper.

We thank the Reviewer for pointing out the lack of clarity in this part of the Introduction. The sentence quoted by the Reviewer was intended to highlight that multi-modal predictive processing is observed in neural circuits beyond

the mammalian cortex and motivate our study on multimodal inputs. The examples that follow, prediction-based pain suppression and motor–visual interactions in cephalopods, were meant to illustrate such cases. In the revised manuscript, we have reframed this section to convey this point more clearly (line 41-42, 46-47). Explicitly applying the results presented in the current manuscript to other brain regions and systems is an active area of work in our group, but goes beyond the scope of the paper. We believe that mentioning the opportunity to apply those results in other systems will help readers appreciate the importance of the findings presented in our paper, towards understanding the circuit mechanisms underlying multi-modal predictive processing across diverse systems.

line 57: "these studies typically focus on predicting a small number of stimuli within a single sensory modality" - studies focusing on visual natural stimuli are hardly using a 'small number of stimuli'

We sincerely apologize for this error in our writing. What we intended to convey is that existing studies have mainly focused on two scenarios: one line of work examines a small number of multi-modal stimulus pairs, while another line of work, as the Reviewer points out, examines high-dimensional (unimodal) visual stimuli. The seminal work on sparse coding and predictive coding of naturalistic images has been particularly inspiring for our study. In the revised manuscript, we have corrected this sentence and now describe the contributions of previous work in a more accurate and appropriate manner (line 59-60).

line 64: "Another major current gap from both experimental and modeling perspectives is predictive processing in high-dimensions" - the original predictive coding paper use visual image patches (200-300 input dimensions). The stimuli used by the authors are much simpler (and lower-dimensional) by comparison.

We agree with the Reviewer that this sentence is imprecise. Our intention was to refer specifically to high-dimensional multi-modal stimuli. We agree with the Reviewer that the original predictive coding paper used high-dimensional image patches. We also wish to clarify that our analytical results for neural representations hold for multi-modal stimuli of *arbitrary* dimensionality, and that our simulation results use 150–400 input dimensions, similar to the dimension of visual image patches in the papers mentioned by the Reviewer. We have revised the manuscript to state this point more precisely (line 67-74).

line 72: "We address these questions by developing a mathematical framework" - not clear what the mathematical framework is. It's certainly not the network model.

We appreciate the Reviewer's comment on this potentially confusing terminology. We have removed this wording entirely in the revised manuscript to prevent further misunderstanding. We have also clarified that the term 'mathematical framework' here specifically refers to the analytical results we derive to characterize the resulting neural representations and circuit structure, rather than the autoencoder network model (line 133-139, 722-726).

We sincerely thank the Reviewer once again for pointing out these important issues in our writing and for helping us sharpen the main messages of our work. We believe the revised manuscript now states our contributions more clearly and accurately.

Figure R2: Comparison between predictive representations in unidirectional and bidirectional predictive processing models. (a) The objective functions of the two models differ. In the bidirectional model, M1 and M3 generate predictions for activity in M2, and the corresponding prediction-errors are minimized (terms highlighted in light and dark blue). This difference in the objective function leads to distinct network architectures. Specifically, the recurrent connections within M1 and M3, as well as the feedforward connections from M1 and M3 to M2, differ between the two models. (b) Throughout training, M3 in the unidirectional model exhibits very weak mismatch responses in the x -only mismatch condition compared to the matched condition. (c) Decoding errors of stimulus- x information based on activity of M3 neurons (i.e., the cross-modal decoding error). We compared the unidirectional and bidirectional models under different stimulus conditions involving 3 stimulus dimensions (indicated in the table above each panel). Cross-modal decoding errors are consistently smaller in the bidirectional model, indicating stronger cross-modal communication across modules. **Left:** decoding between conditions where only x_1 or only x_2 is presented. **Middle:** same as left, but with stimulus y_1 also presented. **Right:** decoding between mismatches along different x dimensions (x_2, x_3 paired with y_1).